# Evaluating the Brexit and COVID-19's influence on the UK economy: A data analysis

**Raghav Gupta[1], Md. Mahadi Hasan[2], Syed Zahurul Islam[3], Tahmina Yasmin[4], Jasim Uddin[1]***

**1** Cardiff School of Technologies, Cardiff Metropolitan University, Cardiff, United Kingdom, **2** Department of Computer Science and Engineering, Asian University of Bangladesh, Ashulia, Dhaka, Bangladesh, **3** Power Integration System, Faculty of Electrical and Electronic Engineering, Universiti Tun Hussein Onn Malaysia, Parit Raja, Johor, Malaysia, **4** School of Geography, Earth and Environmental Sciences, The Institute for Global Innovation, University of Birmingham, Birmingham, United Kingdom

* juddin@cardiffmet.ac.uk

**Data Availability Statement:** All relevant data are within the paper and its Supporting information files.

**Funding:** NO.

## Abstract

The economic landscape of the United Kingdom has been significantly shaped by the intertwined issues of Brexit, COVID-19, and their interconnected impacts. Despite the country's robust and diverse economy, the disruptions caused by Brexit and the COVID-19 pandemic have created uncertainty and upheaval for both businesses and individuals. Recognizing the magnitude of these challenges, academic literature has directed its attention toward conducting immediate research in this crucial area. This study sets out to investigate key economic factors that have influenced various sectors of the UK economy and have broader economic implications within the context of Brexit and COVID-19. The factors under scrutiny include the unemployment rate, GDP index, earnings, and trade. To accomplish this, a range of data analysis tools and techniques were employed, including the Box-Jenkins method, neural network modeling, Google Trend analysis, and Twitter-sentiment analysis. The analysis encompassed different periods: pre-Brexit (2011-2016), Brexit (2016-2020), the COVID-19 period, and post-Brexit (2020-2021). The findings of the analysis offer intriguing insights spanning the past decade. For instance, the unemployment rate displayed a downward trend until 2020 but experienced a spike in 2021, persisting for a six-month period. Meanwhile, total earnings per week exhibited a gradual increase over time, and the GDP index demonstrated an upward trajectory until 2020 but declined during the COVID-19 period. Notably, trade experienced the most significant decline following both Brexit and the COVID-19 pandemic. Furthermore, the impact of these events exhibited variations across the UK's four regions and twelve industries. Wales and Northern Ireland emerged as the regions most affected by Brexit and COVID-19, with industries such as accommodation, construction, and wholesale trade particularly impacted in terms of earnings and employment levels. Conversely, industries such as finance, science, and health demonstrated an increased contribution to the UK's total GDP in the post-Brexit period, indicating some positive outcomes. It is worth highlighting that the impact of these economic factors was more pronounced on men than on women. Among all the variables analyzed, trade suffered the most severe consequences in the UK. By early 2021, the macroeconomic situation in the country was characterized by a simple dynamic: economic demand rebounded at a faster

**Competing interests:** The authors have declared that no competing interests exist.

pace than supply, leading to shortages, bottlenecks, and inflation. The findings of this research carry significant value for the UK government and businesses, empowering them to adapt and innovate based on forecasts to navigate the challenges posed by Brexit and COVID-19. By doing so, they can promote long-term economic growth and effectively address the disruptions caused by these interrelated issues.

## 1 Introduction

Over the past decade, the United Kingdom (UK) has undergone significant transformations that have shaped its political and economic landscape. Two major events stand out during this period: the decision to pursue Brexit and the emergence of the COVID-19 pandemic. Brexit, which concluded on January 31, 2020, marked the end of the UK's longstanding 47-year alliance with the European Union [1]. Concurrently, the UK was confronted with the first cases of COVID-19 within its borders, and by March 2020, the pandemic had swept across the globe, causing widespread fear and discontent. In response to the COVID-19 outbreak, the UK, like many other affected countries, implemented stringent measures, including social and economic restrictions such as lockdowns. While these measures aimed to curb the spread of the virus, they came at a significant cost to both society and the economy [2]. The convergence of Brexit and COVID-19 has raised concerns among experts, who predict substantial economic consequences for the UK, especially during the post-Brexit transitional phase and negotiations with the EU. The combined impact of Brexit and the COVID-19 pandemic is expected to reverberate across various regions and industries within the UK [3]. Therefore, it is crucial for the British people to have access to accurate information about the economic ramifications of these events. The prevailing uncertainty has created unease among investors, resulting in a decline in consumer confidence. Additionally, the British currency, the pound, is anticipated to face significant depreciation in the near future, further exacerbating the existing economic challenges.

In terms of long-run effects, 'Brexiteers' argue that leaving the EU will benefit Britain, while 'Remain' advocates warn of significant economic harm [4]. This is due to the fact that, by leaving the EU, the UK would, on the one hand, have completed decision-making over its future laws and policies, as well as the ability to make its own decisions, so opening up countless growth opportunities. However, when firms experience challenges dealing with the EU, such as border delays and high administrative costs which may result in more unemployment/relocation of personnel, a drop in trade/investment, and, as a result, lower profitability. This will lead to an increase in costs and prices, while competitiveness falls [5]. Further, COVID-19 seems to have the possibility to have far-reaching economic and structural consequences for the United Kingdom economy and its workforce. Recently, the indirect costs of lessening and suppressing the pandemic have risen, and with the beginning of COVID-19, existing trends have accelerated [6], for example, the shift towards online shopping to a great extent and the emergence of more employees who work from home [7, 8]. Moreover, because of national-lockdowns and travel restrictions imposed by COVID-19; the UK's domestic and international commerce has been negatively affected, which has further impacted numerous firms and organisations.

As a result, both Brexit and COVID-19 possibly cause a long-term re-organisation of the UK economy, with continuing ramifications, which might be felt in upcoming years. Due to these discrepancies in forecasts, a detailed examination is required to establish the absolute economic consequence of these events on the UK. Several economists have already

investigated the economic impact of Brexit and the COVID-19 outbreak in the UK. The analysis demonstrates that the consequences currently have been negative since the UK's overall growth, share prices, trade, and exchange rates have mostly deteriorated. Regional disparity is predicted to be exacerbated by the impacts of COVID-19 and Brexit on the economy. The most vulnerable places are London, the Northeast, Wales, the Southeast, and the West Midlands, as these coastal communities depend on tourism and cities are reliant on hospitality for income [9]. In addition, researchers have used sentiment and trend analysis to examine public reaction in the UK and other impacted nations [10]. The results suggest that the majority of the public is concerned about the economic consequences of both Brexit and COVID-19 in the UK. According to the existing studies, the higher government authorities should endeavour to help businesses adapt to new trading agreements and keep border costs as low as feasible. However, the impact of these events on the UK has yet to be documented. COVID-19's economic impact is unlikely to be felt in the same areas and industries that are most vulnerable to Brexit. As a result, this research builds on previous research by evaluating the disparities in the consequences of recent events among UK regions and industries, as well as the evolution of the UK economy over the last decade (2011–2021).

This study aims to explore the changes in the UK economy in three key areas: growth, the people, and trade, as a result of the occurrences of COVID-19 and Brexit. To assess the impact of these events, various data analysis tools and techniques, including the Box-Jenkins method, neural-network modeling, Google Trend analysis, and Twitter-sentiment analysis, were employed. These methods were used to predict and forecast economic factors such as unemployment, gross domestic product (GDP), earnings, and trade in the UK.

The primary objective of this study is to compare the economic levels of the UK before and after the financial crisis. Additionally, the study analyzes and visualizes economic data by region and industry in the UK to identify differences in the impacts throughout the country. Data analytic models were developed and assessed using Python to gain insights into these variations. Furthermore, this study attempts to predict the future consequences of economic factors such as unemployment, GDP, earnings, and trade. Through social media sentiment analysis and trend analysis, the study also investigates the reactions of the British public to these situations.

This study provides valuable information on how the UK economy has changed and how individuals have responded in the context of Brexit and COVID-19. The research emphasizes the need to consider specific economic factors to control for multicollinearity and endogenous variables. Multicollinearity occurs when independent variables are highly correlated in a multiple regression equation, which can undermine the statistical significance of an independent

variable. Therefore, the study focuses on economic indicators that are less influenced by external causes to ensure robust analysis. While other factors could have been included, the study chose to examine key economic indicators that are not significantly affected by external factors. However, future research could explore additional factors to generate a more comprehensive analysis.

Brexit has had a substantial impact on the UK's trade with the EU, which was previously the country's largest trading partner. Exiting the EU required the UK to renegotiate its trade agreements with the bloc, resulting in increased tariffs and trade barriers. As a consequence, trade volume between the UK and the EU has declined, with certain industries, such as the automotive sector, being particularly affected. Moreover, Brexit has influenced investment in the UK, leading to many companies relocating their operations to other EU countries. This has resulted in job losses and reduced economic growth within the country. However, the changing relationship with the EU has also presented opportunities for investment in sectors like technology and finance. The impact of Brexit on employment in the UK has been mixed, with some industries, like finance, experiencing job losses while others, like agriculture, have witnessed increased employment opportunities. The UK's ability to attract foreign workers has also been affected, contributing to skills shortages in specific sectors. Overall, Brexit has had a significant negative impact on the UK's economy, with certain sectors being hit harder than others. While there have been opportunities for growth in select industries, the overall effect of Brexit on the UK's economy has been detrimental, resulting in reduced trade, investment, and economic growth.

The following paper is structured as follows: In section 2 highlighted the existing literature and the consequence of Brexit and COVID-19 in the UK economy. In section 3 displayed the different data analysis, models and various methods are described in theoretically for a better understanding of the data. In section 4 presents the detailed parameters of various results section that shows the data visualisation. In conclusion, there will be a brief discussion of how these findings may affect the outcome and further point to the limitations and future work.

## 2 Background study

The UK's European Union (EU) membership was decided through a June 23, 2016, referendum, with 51.9% voting to leave out of 33,551,983 total votes. This outcome has caused uncertainty in the tourism sector, impacting consumers, industry players, and policymakers [11]. In the de-internationalization perspective [12], it was often perceived as a failure, discouraging firms from limiting or discontinuing exports to foreign markets. Definitions of de-internationalization were mostly limited, but some comprehensive ones acknowledge its partial or complete nature. However, they often associate it with voluntariness rather than a strategic response to changes in the global business environment. The lack of transparency in formal policies and their impact creates a growing risk for international business [13]. This risk manifests not only through partial de-internationalization, such as decoupling, but also through complete withdrawal and exit from the international market. Compliance with minimal regulatory requirements poses a significant threat, leading firms to strategically opt for de-internationalization as a means to limit their involvement in international operations.

Together, with the rising importance of research in the economic repercussions of COVID-19 as a global concern and Brexit, the experimental investigations of the economic factors (unemployment, GDP, earnings and trade) of these incidents are limited. Campos et al. [14] examine the effect on migration and trade between the UK and the EU due to Brexit and utilised a structural gravity model. Further undertaking a quantitative analysis, the authors shows that there have been severe repercussions for the UK's trade and migratory patterns. Kaminska

et al. [15] signifies published documents on the consequences of Brexit, such as the House of Commons Treasury Committee (2018), HM Treasury (2016) (Crafts, 2016) and the Bank of England (2018). Despite each of these published papers adopting different scenarios and techniques in analysing the consequences, their findings were common and indicate undesirable effects on the UK economy. The IMF (2018) estimated that production will fall by 2%—8% as a result of Brexit. Their analysis includes a simple calculable general equilibrium model and a number of assumptions about the relative sizes of various Brexit transmission channels (International Monetary Fund, 2018). In another study, a cross-section of pre- and post-Brexit surveys were conducted to look at the industrial development (IDV) nexus in the UK economy [16]. The purpose of this research was to look at the primary drivers of industrial development in the UK before and after the implementation of a financial reform plan. Researchers employed multiple regression analysis to anticipate changes in the dependent variable (IDV) induced by independent factors (Brexit). Later, shocks were described, quantified, and explained using the impulse response function (IRF), vector auto-regression (VAR), and variance decomposition. The data demonstrated that all independent factors, trade openness, equity openness, and capital account openness were significant predictors of commercial growth in pre-Brexit polls. However, regression analysis shows that only equity openness (EO) and trade openness (TO) had a significant influence on Post-Brexit Polling Industrial Development (IDV) at the 5% level of significance. As a result, following Brexit, the authors recommended that the UK should rethink important financial policies in order to boost future industrial growth [16].

The interdependence of economic and public health has become evident during the pandemic [17]. It highlighted that a nation's economic well-being relies on the health of its citizens, while also emphasizing the global significance of people's well-being. The imposition of lockdown measures worldwide demonstrated the need to pause economies to control the spread of SARS-CoV-2. Although economic health metrics like consumer price index (CPI), gross domestic product (GDP), and unemployment rate are well-established, public health measures can be enhanced by considering contemporary indicators such as emotional well-being and mental health. These measures can have significant implications for human capital and productivity in the economy, especially during times of crisis like the pandemic.

Unlike past crises such as aviation tragedies, natural disasters, and supranational union exits [18], the COVID-19 pandemic has had an exceptional impact on the tourism and gambling sectors. Unprecedentedly, widespread lockdowns were implemented in cities and countries, an unprecedented measure. While these actions are vital for saving lives and demand global cooperation to contain infectious outbreaks like COVID-19, they also pose a substantial employment risk, particularly in tourism-dependent cities where it constitutes a primary source of income.

Researchers have examined the impact of Brexit using a variety of data analytic approaches. In order to analyse the Sterling Pound's predictability, [19], used Google Trends data from the last five years to do a prediction analysis on the Pound's exchange rates to Euro and Dollar. The goal of the study is to determine if the pound and Google query data are connected by examining the relationship between the pound and Google query data on 'Pound' keywords and subjects from the 2016 UK referendum through January 31st, 2020. The findings reveal that there are statistically significant quantile correlations between Google query data and pound exchange rates, pointing to one of the field's most important implications: detecting whether changes in one economic measure elicit reactions in other economic measures.

Simionescu et al. [20] also look at how Brexit has affected the monthly unemployment rate since the referendum. This is one of the most important indicators of the country's long-term development. This study stands out because it uses microdata to illustrate the political

uncertainty caused by Brexit, with Google Trends serving as the data-collecting tool. Statistics for the four countries that make up the United Kingdom (Wales, Northern Ireland, England, and Scotland) are also analysed using a multilayer and panel data structure. Despite the paucity of data to back this assertion, the findings are consistent with an analysis of important macro-economic indicators, demonstrating that uncertainty caused by Brexit lowered unemployment from June 2016 to March 2019. The study suggests that government policies should stimulate investment in order to aid the UK's future financial expansion and growth. On the other hand, Keogh-Brown et al., [7] assess COVID-19's economic impact in the UK by considering direct illness consequences, public health interventions, and policies. COVID-19, according to their findings, may have a £39.6 billion economic impact on the UK (1.73 percent of GDP). The data reveal that COVID-19 has the potential to wreak havoc on the UK economy, and while the government attempts to reduce mortality appear to be vital; the length of the school and corporate closures is crucial in deciding the economic impact. Finally, the study suggests that the UK government's first economic assistance measures may need to be enhanced if the pandemic is to be adequately handled without triggering the collapse of many enterprises and the loss of many workers' jobs. Furthermore, researchers wanted to know how people felt about the COVID-19 vaccination before it was released in the United States and the United Kingdom.

The COVID-19 pandemic led to a global quarantine in 2020, affecting economies worldwide, including Malaysia [21]. The quarantine measures aimed at controlling the public health crisis had mixed effects on economies. The agricultural sector faced significant challenges due to movement restrictions, resulting in disruptions in the farm-to-consumer supply chain. Travel limitations had a notable impact on food delivery and all stages of agricultural production. Although panic buying has diminished and logistical issues for fresh produce are being addressed, there are concerns about the country's ability to maintain food self-sufficiency during future emergencies and potential trade disruptions.

Externalities are external events or shocks that can influence the internal operations of organizations and industries [22]. The global impact of the COVID-19 pandemic exemplifies such an externality, affecting a wide range of organizations and industries worldwide. Externalities often drive research and strategic discussions among academics, professionals, and policymakers. However, it is important to recognize the limitations of descriptive and association-oriented research in accurately predicting and prescribing causal effects, as they may lack essential factors like boundary conditions. Drawing causal conclusions from non-causal research designs is inappropriate.

According to another study by Leppeman et al. [10], between July 28 and August 28, 2020, COVID-19 vaccine-related social media posts from the US and the UK were analysed using neural linguistic programming with human validation. The sentiment analysis investigated the polarity of the comments (positive, neutral, and negative) as well as the themes that appeared in the negative ones. The US and the UK had a net sentiment profile of around 8% negative, 28% positive, and 63% neutral in 243,883 social media posts. Further data revealed that sentiments about COVID-19 vaccinations on social media in the two countries varied significantly.

There were detected variations in negative emotion themes. Negative sentiments in the US stemmed mostly from health and safety concerns, the fear of obligatory vaccination, and the role of pharmaceutical firms in vaccine distribution. In the UK, the major cause of criticism was the concern about mandating vaccination (almost doubling the amount of vaccine). In the third quarter of 2020, the UK exhibited widespread opposition to COVID-19 vaccinations. According to the authors, authorities in both nations may use the reasons for unpleasant emotions to develop evidence-based initiatives to counteract COVID-19 vaccine rejection. Knipe et al. [23] also utilised Google Trends data (from January 1, 2020, to June 9, 2020) to examine

the shifting resident anxiety trend in the UK in response to the government operations, as measured by fluctuations in the search frequency for mental agony, coping, and resilience. During the outbreak, it analysed how particular themes changed in connection to significant dates, as well as the most popular phrases. COVID-19 was no longer related, either directly or indirectly, to the inquiries with the most significant rise over time.

The global COVID-19 pandemic has not only caused a public health crisis resulting in loss of life and widespread suffering but has also severely strained economies worldwide [24]. Lockdown measures were implemented across the globe, including in Malaysia, leading to a temporary halt in economic activities and subsequent reorganization. This unprecedented phenomenon is conceptualized in this chapter as the "quarantine economy." Although the initial pause was brief, governments gradually reopened their economies while implementing new social practices such as remote work, physical distancing, and visitor records. These measures facilitated the adaptation of economic activities to the ongoing pandemic situation, resulting in a reconfiguration of how businesses operate.

In regard to the Business to Business (B2B) marketing [25], it has increased focus on marketing issues and their implications in the increasingly turbulent B2B market. It sheds light on B2B marketing strategies that are relevant during major crises, employing the marketing mix as an organizing framework and utilizing the COVID-19 pandemic as a natural context for transformative marketing.

Specifically in the transformative marketing emphasizes "competition" and "superior value," which may contradict the goal of delivering "benefits to all stakeholders" due to potential casualties (e.g., dropouts from competition) or costs (e.g., hidden sacrifices or tradeoffs) associated with such focus. However, the lessons learned from collective efforts in combating the challenges of COVID-19 suggest that "collaboration" and "shared prosperity" are crucial for leveraging limited resources and providing agile and valuable responses. These lessons ensure that no one is left behind during times of adversity.

The digital economy and advancements in technology have made global business an ever-present force [26]. Firms now have the ability to automate processes, target customers, and access larger global markets. The COVID-19 pandemic has further emphasized the importance of global business, enabling firms not only to survive but also to thrive, contributing to the overall functioning of the economy and society.

What distinguishes the current period from past global crises is the remarkable speed at which firms have shifted from survival mode to pursuing growth and success. Industries like cleaning, delivery, and technology services have showcased exceptional adaptability. The unprecedented changes brought about by the pandemic have created a new normal, demanding the exploration of new ideas and a reassessment of existing ones. This is vital for forging a transformative path forward and attaining organizational excellence in the realm of global business.

In the latest study, Sharma et al. [27] addressed both the public's concerns and the expanding search trend as the number of COVID-19 cases increases in important nations. Statistics from eight major nations (Spain, China, the US, Italy, the UK, India, Iran, and France) were collected for their analysis. In these eight countries, the Google search Trend for "COVID-19" was analysed. The expanding Google Trend reflected the public's sadness, anguish, and fear in response to the epidemic. From March 10th to April 10th, 2020, the trend has grown significantly. Throughout the observation period, the Google interest wave in the examined nations demonstrated a sequence of high values. The average interest amount for Google has been determined for two additional periods: (01 January to 29 February) and (01st March to 10th April). This indicates that public response has increased over time. According to the study, Google Trends may be utilised to discover which regions of the country are least and most

affected. Numerous additional researchers, like Chandio and Sah [28] and del Gobbo et al. [29], have utilised Twitter sentiment analysis, whilst Garcia and Berton [30] and Georgiadou et al. [31] integrated big data analysis with sentiment analysis and event study to capture the global effect of Brexit and COVID-19.

## 3 Methods

The study has been classified into three essential areas: development, exchange, and individual (person) of the United Kingdom. It examines how the combined catastrophes affected the UK economy and how individuals responded by utilising industry and region-specific data. Numerous data analysis methods and methodologies, such as neural networks and Google Trend Analysis have been utilized for investigating the future economic variables of the United Kingdom in order to measure the effects of both Brexit and COVID.

This research obtained data extraction procedure and the methodology utilised for the analysis and to investigate the economic trajectory and public reaction in the United Kingdom before and after Brexit and COVID-19. For the study of Brexit and COVID-19, the study has collected the required information (data) for the past decade from 2011 until 2021. This study investigates the differences between Pre-Brexit, Brexit, Post-Brexit, and COVID-19. The required data has been collected from reputable web sources the Office for National Statistics (ONS), Google Trends, and Twitter. ONS is the UK Government website that provides open and transparent access to UK economic statistics. The Office for National Statistics (ONS) is dedicated to being open and transparent about the data they hold. It also shows where that data has come from and gives the main uses of that data. Various data visualisation and data analysis technologies are utilized to examine the obtained data that helps to investigate the impact of Brexit and COVID-19 among the industry sector, and its people of the United Kingdom. In addition, to validate the dataset and forecast the effects of the UK economic conditions on unemployment, GDP, earnings, and trade. The extracting data has been analysed and measured using statistical numerical software. For data forecasting validation and predictive analysis, a neural network model has been proposed using Python.

**Explicit Statement**: In light of the economic trend and public reaction across the United Kingdom, this study is classified into three key stages: 1) before; 2) during; and 3) after Brexit and COVID-19, throughout the years between 2011–2021. The extracted data (ONS, Google Trends, and Twitter) was then analysed using relevant statistical software tools to facilitate data conversion and measurement comparisons. Python has been employed to construct a neural network model to perform data validation, and the Box-Jenkins method was implemented for forecasting and predictive analysis. In addition, this study also evaluated Twitter sentiment using the Twitter API and the Text-Blob method in Python. The relevant studies were identified from entirely open sources that fully complied with the terms and conditions.

### 3.1 Data analysis

The information on the UK's economic variables originates from the National Statistics sources (Office of National Statistics, 2020), which allows a comprehensive investigation of projected changes across different economic outcomes, regions, and businesses in the UK as a result of Brexit and COVID-19. The data obtained from ONS is a time series of quarterly data (42 quarters) over the past ten years (2011–2021) and encompasses among each the instances under analysis, Brexit and COVID-19. This study investigates three different time periods between 2011 and 2021 for comparing the UK economy prior to, during and following the situations such as "Pre-Brexit" [Q1 (January to March) 2011 to Q2 (April to June) 2016]; "Brexit"

[Q3 (July to September) 2016 to Q4 (October to December) 2019] and, Post-Brexit & COVID-19 [Q1(January-March)—Q2 (April-Jun), 2021]. The data has been gathered based on the four most important economic factors: "Unemployment," "Gross Domestic Product (GDP) index," "Earnings," and "Trade." In addition, the data collection was categorized into four regions namely—England, Scotland, Wales, and North Ireland and the twelve industries including Agriculture, manufacturing, Construction, Wholesale, Transport, Accommodation, Financial, Science, Public Admin, Education, Health, and Others. In addition, to represent the impact of the British populace, gender-specific economic data is produced (men and women).

The unemployment rate varies between 3.8% to 8.4%, with an average of 5.65%, as shown in Table 1. Additionally, the GDP has varied from 73.3% while 102.5%, average weekly earnings are £595.05, and average net trade stands at 10140.88 million pounds. The table also shows the Pre-Brexit conditions were the most adverse on average, with the largest unemployment rate, the lowest GDP growth rate, and the lowest incomes. On the other hand, during the Brexit transition phase, the economy improved, but net trade was severely impacted. In the post-Brexit and COVID-19 period, the economy initially endured severe economic catastrophe, but subsequently started to recover when GDP and profits began to climb. This demonstrates that the UK has experienced a transition over the last decade This study begins with a graphical depiction of the quantitative data using line and bar graphs for each economic consequence utilising various Microsoft Excel and Python tools.

**3.1.1 Neural network model.** In this section, a neural network regression model has been developed. It has been used in different disciplines and the regression analysis applied to the extent of the relationship between a dependent variable and a set of independent variables [32].

The neural network (NN) is a substantial deep learning technique as it can be trained to estimate the parameters and incorporate the behaviour provided by the samples. The NN model consists of an activation functional as well as a network structure with an input layer, several hidden layers, and an output layer. Based on the significance of each neuron input, the functions determine whether or not each neuron should be triggered for predictions. The output of a neuron is the product of the input value times the weight, which is then sent to the next layer [33]. Deep learning neural network models utilize nonlinear activation functions, which are crucial for learning and modeling complicated data [34]. Deep learning neural networks may reuse characteristics gained in one hidden layer in subsequent hidden layers; hence, NN modeling was utilised for this investigation. This enables a deep neural network to measure performance and imitate a range of natural tasks with a limited number of weights and units [35]. This method facilitates the generation of improved regression results. Python is used to create the NN model using the UK economic dataset collected earlier.

**Table 1. Summarizes the descriptive statistics of the four main UK's economic factors analysed in this paper in all the 42 quarters between the years 2011–2021.**

| Model | MCC | Cohen's kappa | Hamming Loss | Precision |
|---|---|---|---|---|
| Count | 42 | 42 | 42 | 42 |
| Mean | 5.65 | 87.97 | 595.05 | -10140.88 |
| Std.Dev | 1.58 | 9.11 | 40.31 | 2064.41 |
| Min. | 3.8 | 73.3 | 540.62 | -16400.66 |
| Median | 5.1 | 87.3 | 585.93 | -10108.5 |
| Max. | 8.4 | 102.5 | 677.001 | -4677 |

## 3.2 Box-Jenkins method

This study examines the Box-Jenkins method, the fundamental and most widespread technique for time series analysis [36, 37]. These univariate models aim to better interpret and predict future observations for a single time-dependent variable. Box Jenkins has been recognized as an attractive choice for data sets that are mostly consistent and have moderate volatility. The fundamental assumption of these models appears the data remain stable. The strategy for detecting outcomes relies on data point variations. Researchers must account for hysteresis and reduce any turbulence and periodicity as feasible from older data sets. This approach enables the model to identify patterns utilising three fundamentals such as autoregression, differencing, and moving average [38].

This study has estimated the Box-Jenkins model using Python. It has been used in three consequence steps, a Box-Jenkins time series model is developed. In this stage, the data and other relevant information are utilised to choose a sub-section of the model that would eventually summarise the data. The software generates the Autocorrelation Function (ACF) graph, which displays the association between lag values and an observation. In addition, a partial autocorrelation function (PACF) graph depicting correlations for observation with lag values utilising the earlier lagged data are created. There are various approaches have been used to examine the stationarity features of time series data [39].

The null hypothesis of the test is that the time series is nonstationary due to the presence of a unit root. The test's p-value is utilized to evaluate the test result [40]. If the p-value is less than the threshold (5% or 1%), the null hypothesis is rejected and the time series is stationary. If the p-value is greater than the threshold, the null hypothesis is not rejected, and the time series has been deemed non-stationary. Kwiatkowski-Phillips Schmidt-Shin (KPSS) is a sort of Unit root test or statistical technique used to determine if a series is stationary around a deterministic trend [41]. To reject the null hypothesis, the test statistic must exceed the basic values provided. The p-value should be low if the desired critical value is genuinely exceeded. The KPSS statistic will exceed the critical threshold of 5 percent if the p-value is less than 0.05. The number of lags shown is the number of lags utilized by the model equation of the KPSS test. Typically, one must define the requisite p, d, and q variables in a basic auto regressive integrated moving average (ARIMA) model. To decrease non-stationarity, it is generated the values using statistical methods by conducting a difference. In Auto ARIMA, the model will determine the ideal p, d, and q parameters for the data set to provide more accurate forecasts. The last stage of the model evaluates the fitted model considering the available data and looks for regions where it may be improved. Two diagnostic factors to consider are overfitting and residual errors. Initially, assessed if the model is overfitting the data. This often indicates that the model appears more complex than it is and that output from the training set has been captured. This is a concern in time series forecasting because it restricts the model's capacity to generalize, resulting in poor out-of-sample prediction and forecast performance. Both in-sample and out-of-sample performance must be considered, necessitating the development of a comprehensive model evaluation test harness. The residual forecasts are a valuable diagnostic tool and the predicted residual time series would lack temporal structure in an ideal model. Furthermore, in an ideal model, the anticipated residual time series would have no temporal structure. There is a serial correlation in the residual errors, indicating that this information may be included in the model.

Following the study of economic data analysis, this study has undertaken a sentiment and trend analysis of social media to obtain a better understanding of how the British public reacted to the consequences of Brexit and COVID-19. According to previous studies, several approaches and procedures, such as Google Trend and Twitter sentiment analysis, are utilised to determine how individuals reacted to the Brexit and COVID-19 events.

## 4 Results

### 4.1 Unemployment rate

This study describes the process here on the key findings of the various analysis conducted to determine the impact of Brexit and COVID-19 on the UK. Initially, the four economic parameters Unemployment, Earnings, GDP, and Trade are examined in relation to the events based on the timeframe from 2011 to 2021 in a particular region, industry, and gender areas. The neural network regression model is evaluated. The Box-Jenkins Method is utilised to analyse the time series data and then anticipate the subsequent years. All the data analysis, exploratory, visualisation were extracted from ons.gov.uk. Finally, social media analysis is used to assess the public's reaction.

The graph in Fig 1 depicts the overall unemployment rate (%) in the UK from 2011 Q1 to 2021 (ONS, 2021). The pattern of the line graphs indicates that the unemployment rate was greater at the beginning of the decade before it began to decline in 2014. (Pre Brexit). The rates decreased steadily until 2019(Brexit) when they suddenly increased in 2020 Q1 (Post- Brexit & COVID-19). By the end of 2020, the UK has taken control of the rate and it began to drop.

Fig 2 is a summary of regional unemployment in the UK. The effects of the two catastrophes varied throughout all regions. During the past decade, England and Scotland had lower unemployment rates than Wales and Northern Ireland. So, between 2016 and 2020 (Brexit), the unemployment rates varied, in the post-Brexit and COVID-19 period, all regions have seen an increase while with England having the highest and Northern Ireland the lowest (2020). By the beginning of 2021, the regions began to exercise control over their labour markets.

Fig 3 illustrates the average regional unemployment rate across gender in the UK during the three periods analyzed in this study. It is notable that the unemployment rate has consistently been higher for men compared to women across all periods in the UK. Moreover, when examining the regions, Wales and Northern Ireland display higher unemployment rates than England and Scotland. However, the impact of Brexit and COVID-19 has had a more significant effect on unemployment in England and Scotland compared to Wales and Northern Ireland.

Fig 4 depicts the average unemployment rate in the UK's major industries Pre-Brexit, Brexit, COVID-19, and post-Brexit. The industries with the highest unemployment rates were

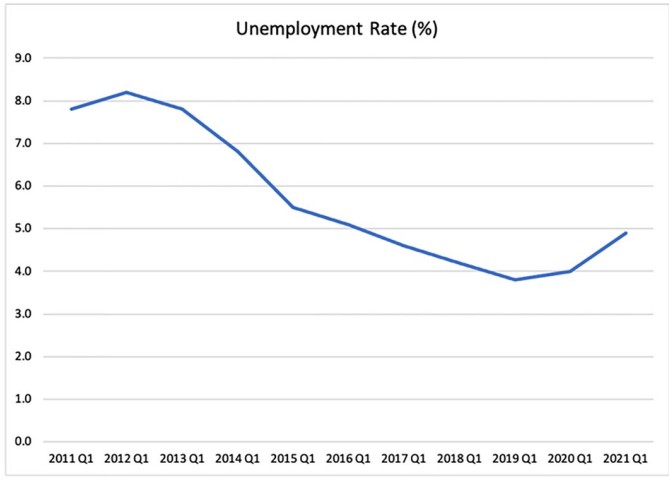

**Fig 1. Unemployment rate in UK (2011–2021).** Source: ONS.

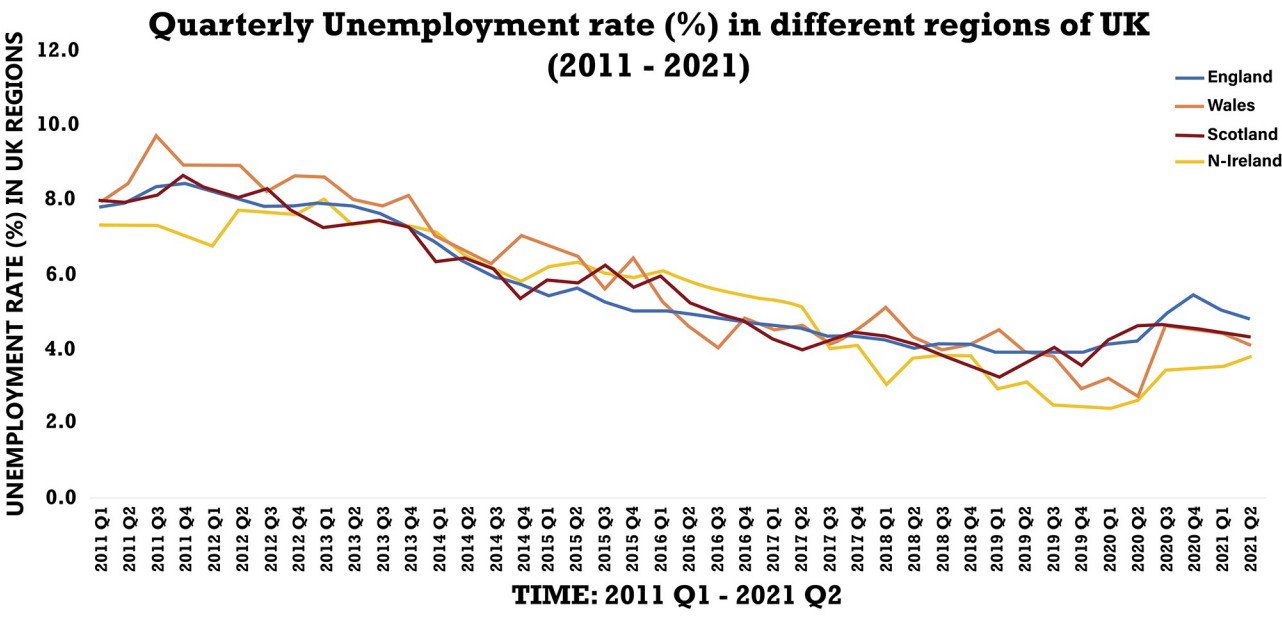

**Fig 2. Unemployment rate in all regions of UK.** Source: ONS.

'Accommodation', 'Transport', 'Construction', 'Wholesale', and 'Other'. During the Brexit period (2016–2020), it is evident that the unemployment rate decreased, but these industries remained at the top. Furthermore, COVID-19 and Post-Brexit had the greatest impact on these and all other industries. Even throughout COVID-19 and the post-Brexit period, industries like Public Administration and the health sector had low unemployment rates.

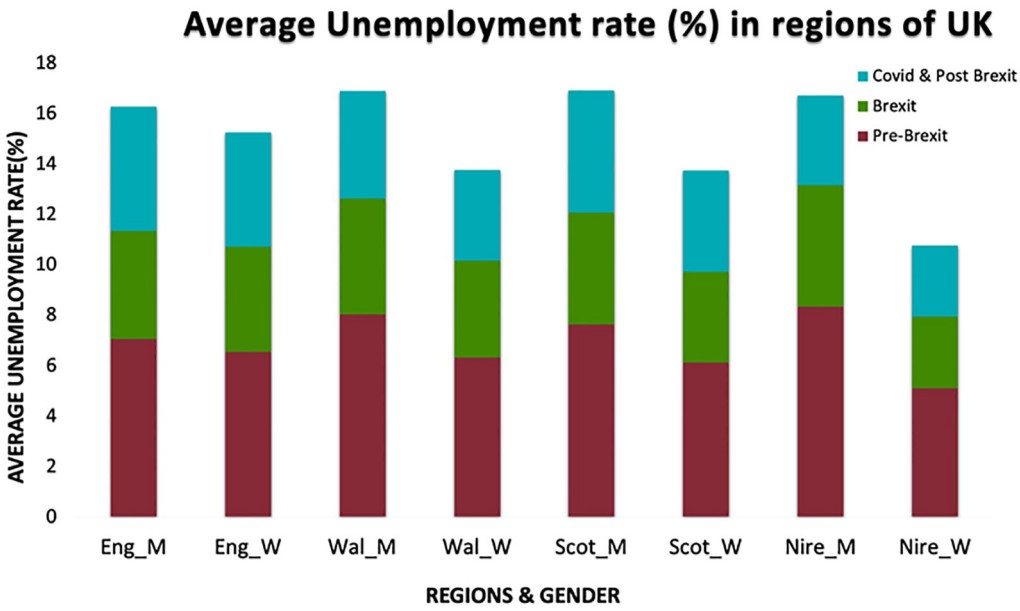

**Fig 3. Unemployment rate in UK gender-wise.** Source: ONS.

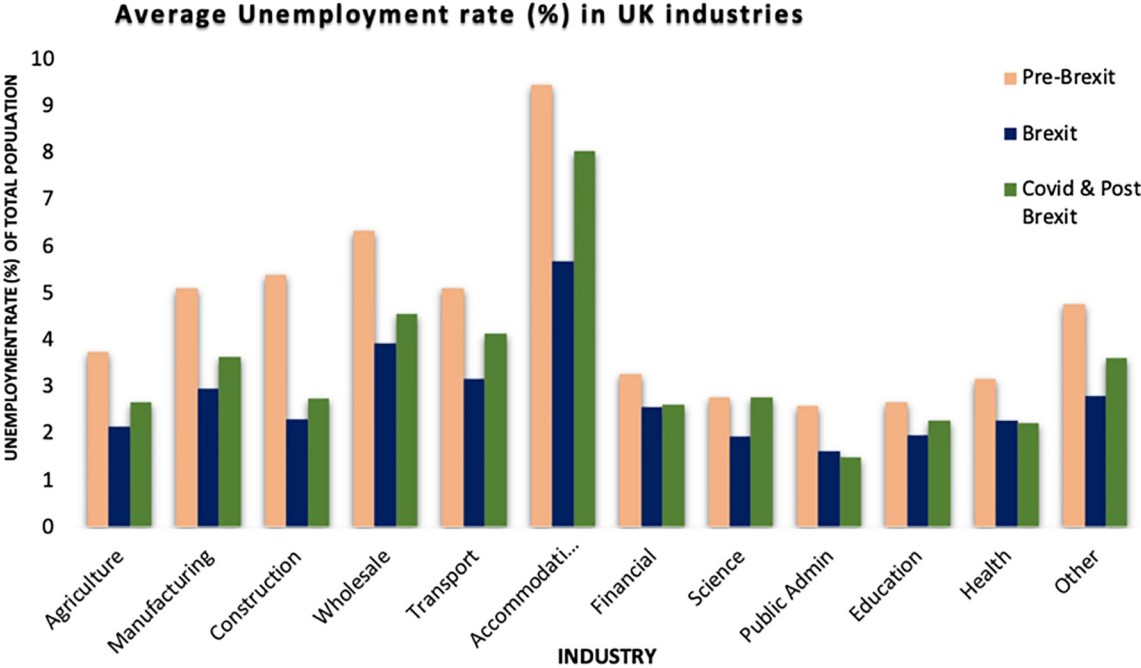

**Fig 4. Unemployment rate of industries in the UK.** Source: ONS.

## 4.2 Earnings (average weekly)

This study further examines the effect of the two events on the UK's earning analysis.

Fig 5 aggregates the total and regional incomes (mean average pound per week) of the UK between 2011 and 2021. The data indicate that England had the greatest incomes, followed by Scotland, Wales, and Northern Ireland. Considering the period, all regions' revenue has increased throughout the Brexit phase. In COVID-19 and post-Brexit periods, incomes for all areas decreased for two quarters before increasing again in 2021, except for England, where wages increased continuously.

Fig 6 provides an overview of the average regional earnings for the Pre-Brexit, Brexit, COVID-19, and Post-Brexit periods. England and Scotland have had more relative wage increases than Wales and Northern Ireland.

Fig 7 indicates the average percentage change in weekly wages across industries in the UK throughout the pre-Brexit, Brexit, COVID-19, and post-Brexit periods. During the Brexit, COVID-19, and post-Brexit periods, the average incomes across all industries increased. 'Science,' 'Financial,' 'Public Administration,' and 'Health' are the industries with the greatest rise in profits. On the other hand, the incomes for businesses such as "Accommodation," "Education," and "Agriculture" were comparatively lower.

## 4.3 Gross Domestic Product (GDP) index

Gross Domestic Product (GDP) serves as a standard measure to assess the value generated through the production of goods and services within a country over a specific period. It reflects the revenue generated by this production and the total expenditure on final products and services, excluding imports. However, GDP alone does not consider the impact of inflation or rising prices, regardless of whether GDP increases or decreases. To address this limitation, the

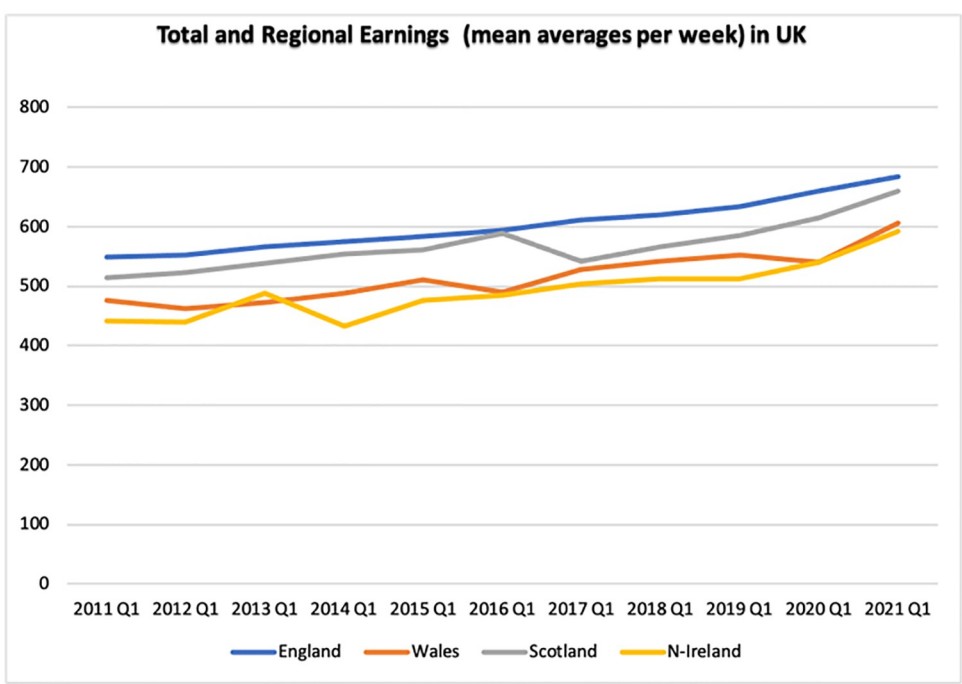

**Fig 5. Mean average earnings per week of the UK and its regions.** Source: ONS.

GDP price deflator, also known as the GDP index, comes into play. It evaluates the influence of price changes on GDP by selecting a base year and comparing current prices to those of the reference period. The GDP price deflator effectively measures the extent to which price fluctuations impact changes in GDP [42]. It tracks the prices paid by businesses, the government, and consumers, providing insights into the variations in price levels or inflation within the

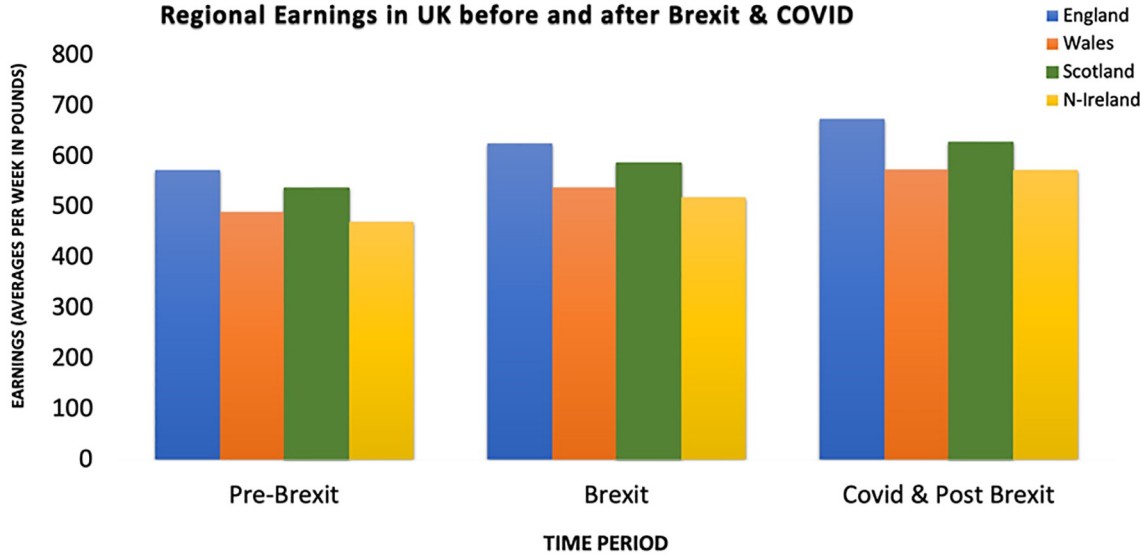

**Fig 6. Comparison between earnings of different regions of the UK.** Source: ONS.

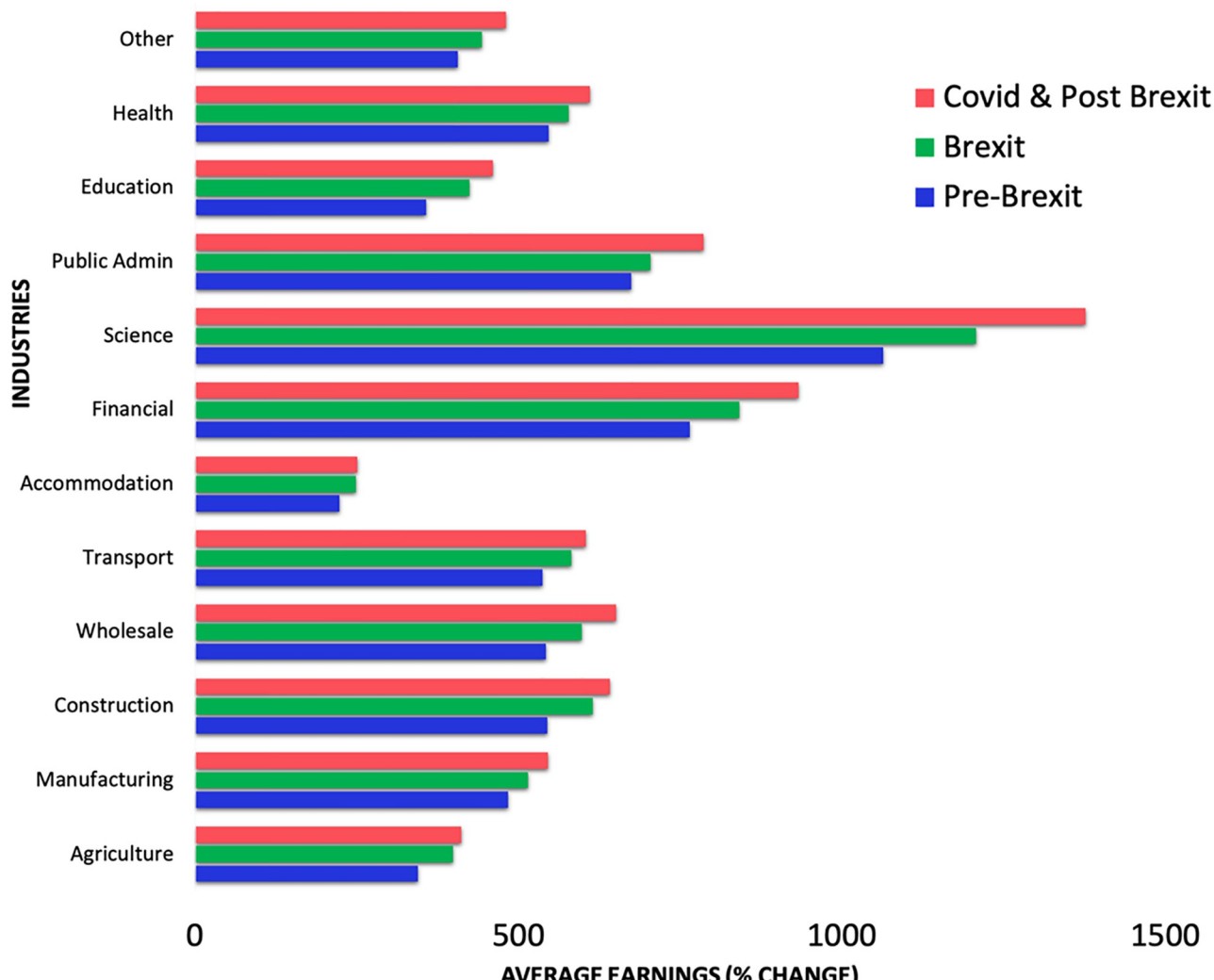

**Fig 7. Average earnings in different industries of UK.** Source: ONS.

economy. By utilizing the GDP price deflator, economists can compare the real volume of economic activity across different years. This comparison is crucial because evaluating the GDP of two years with distinct price levels can lead to misleading conclusions [42]. Therefore, the GDP price deflator enables a more accurate analysis of economic activity by considering the impact of changing price levels over time.

Fig 8 depicts the UK's GDP price deflator index from 2011 to 2021. It may be interpreted that the GDP increased modestly until 2020, then declined for two quarters during COVID-19 and post-Brexit before beginning to rise again in 2021.

Fig 9 displays the average GDP index of the various UK industries throughout the pre-Brexit, Brexit, and post-Brexit periods. It is proven that the contribution of the industries grew during the Pre-Brexit, Brexit period and decreased during the COVID-19 and Post-Brexit and

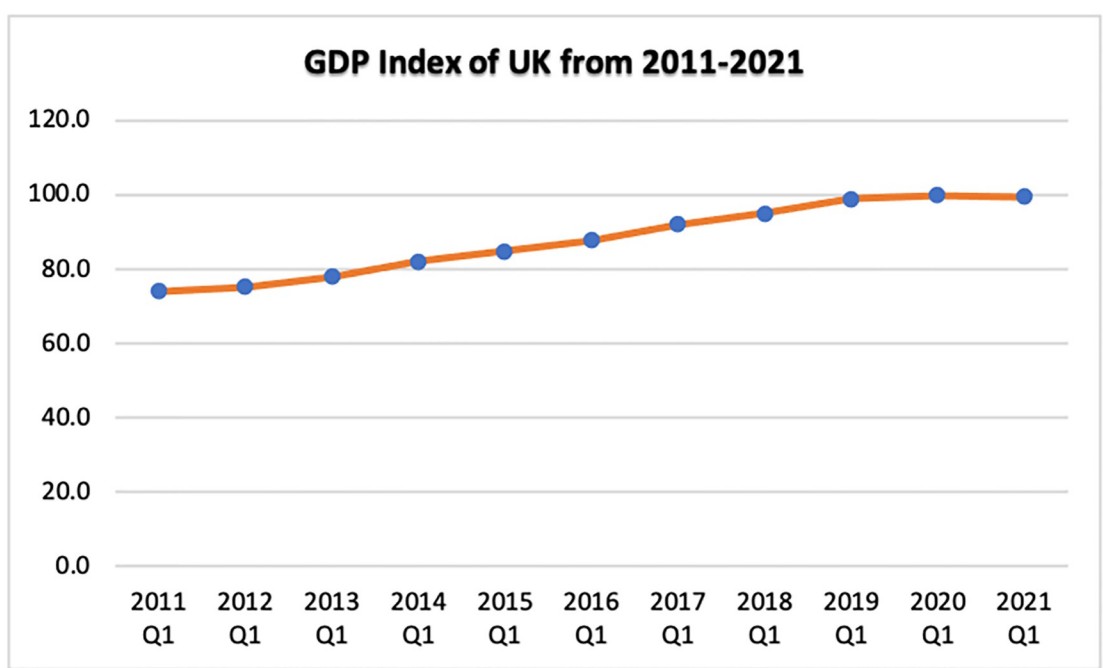

**Fig 8. GDP index of the UK for the last decade.** Source: ONS.

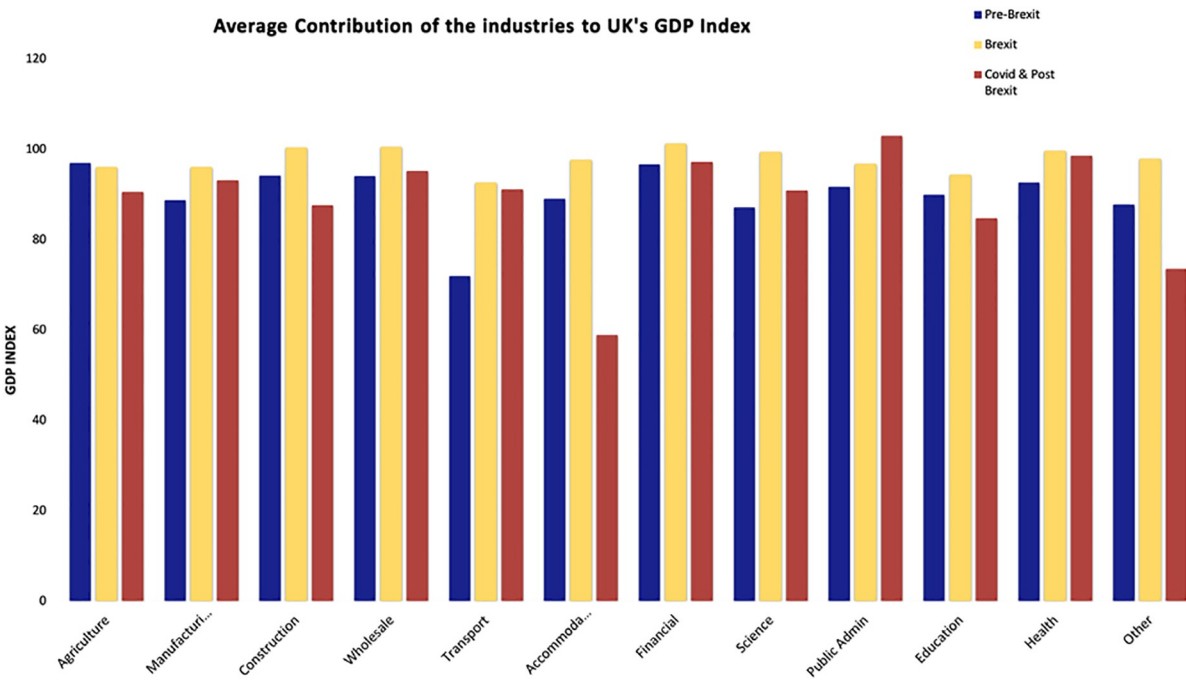

**Fig 9. The average contribution of the industries to UK's GDP index.** Source: ONS.

Brexit periods. During the Post-Brexit and COVID-19 period, the accommodation, construction, education, and agriculture industries experienced the most from COVID-19, while the science, financial, public administration, and health industries contributed the most to the UK's GDP.

## 4.4 Trade

Since the UK has been a worldwide trading force, it is essential to evaluate the effects of Brexit and COVID-19 on UK trade.

The commerce, imports, and exports of the UK from 2011 to 2021 are depicted in Fig 10. Prior to 2019, it is evident that the UK had a negative overall trade value with imports exceeding exports. The trade situation improved in 2019–2020 as total exports rose and total imports dropped, putting the overall trade value near zero/positive. Beginning in 2021 in COVID-19 and post-Brexit period, the total imports increased while total exports dropped, resulting in a trade imbalance.

In Fig 11 provides a summary of the UK's trade relations with European Union (EU) nations and non-EU countries, individually. In comparison to the UK's trade with EU nations, the Non-EU trade has increased and has been greater during the past decade. In addition, with Brexit and COVID-19, non-EU trade has increased relative to EU trade, but overall, the trade has dropped due to COVID-19.

Fig 12 depicts the average volume of trade in various regions of the UK throughout the pre-Brexit, Brexit, and post-Brexit and COVID-19 periods. England is the only country having the largest trade imbalance. Other regions, like Scotland, Wales, and Northern Ireland, have positive trade that is close to zero. During Brexit, trade levels increased in Scotland and Northern Ireland, but still, the trade imbalance widened in England, and trade levels declined but remained positive in Wales. During the Post-Brexit and COVID-19, Scotland, Wales, and Northern Ireland had a decline in trade, but England's trade deficit was reduced.

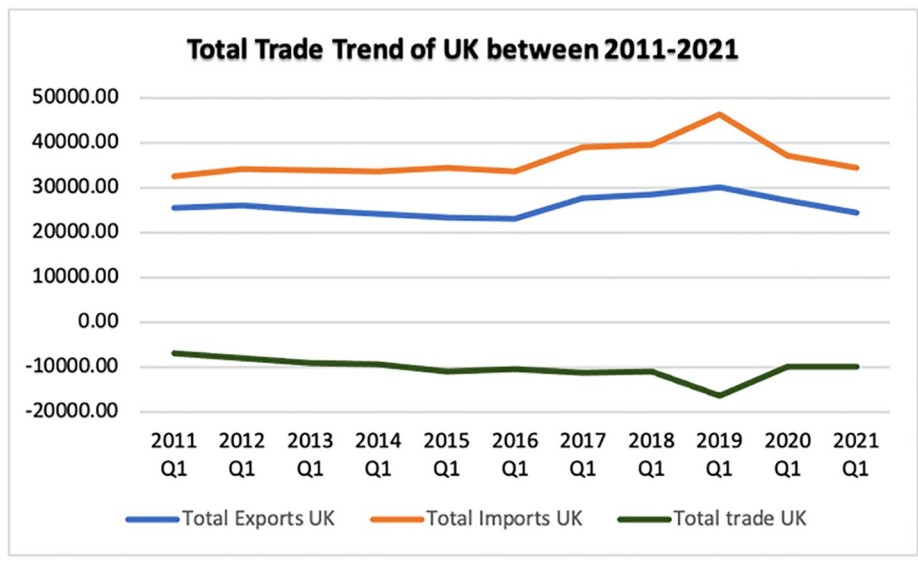

**Fig 10. Total trade of the UK in the last decade.** Source: ONS.

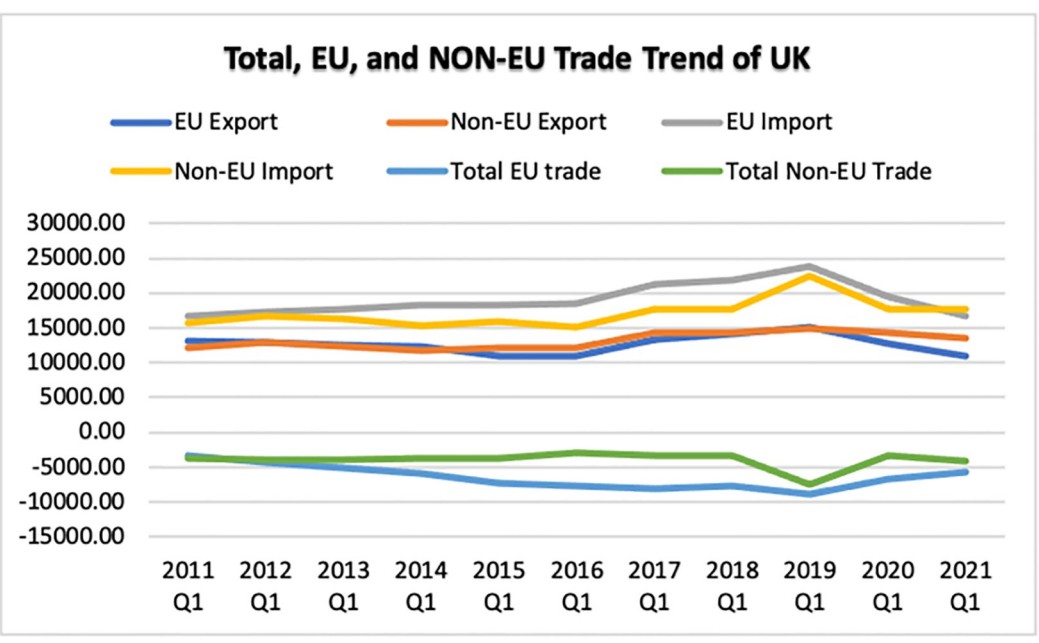

**Fig 11. UK trade with EU and Non-EU.** Source: ONS.

## 4.5 Neural-network model

Furthermore, utilizing the UK's economic statistics, this study developed a neural network regression model and evaluated the appropriate findings. To improve the regression model, hyperparameter tuning is recommended to use a batch size of 20, a number of epochs of 200, and an optimizer input of RMSProp. Using the aforementioned parameters in the regression model, the training and validation loss was computed. Originally, the difference between the

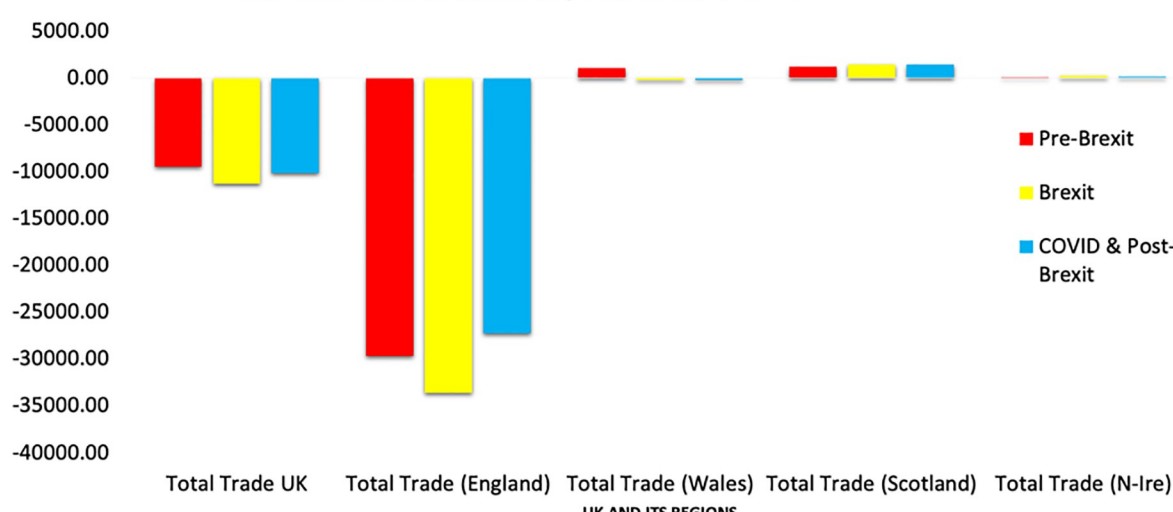

**Fig 12. Comparison of the UK's total trade in three different periods.** Source: ONS.

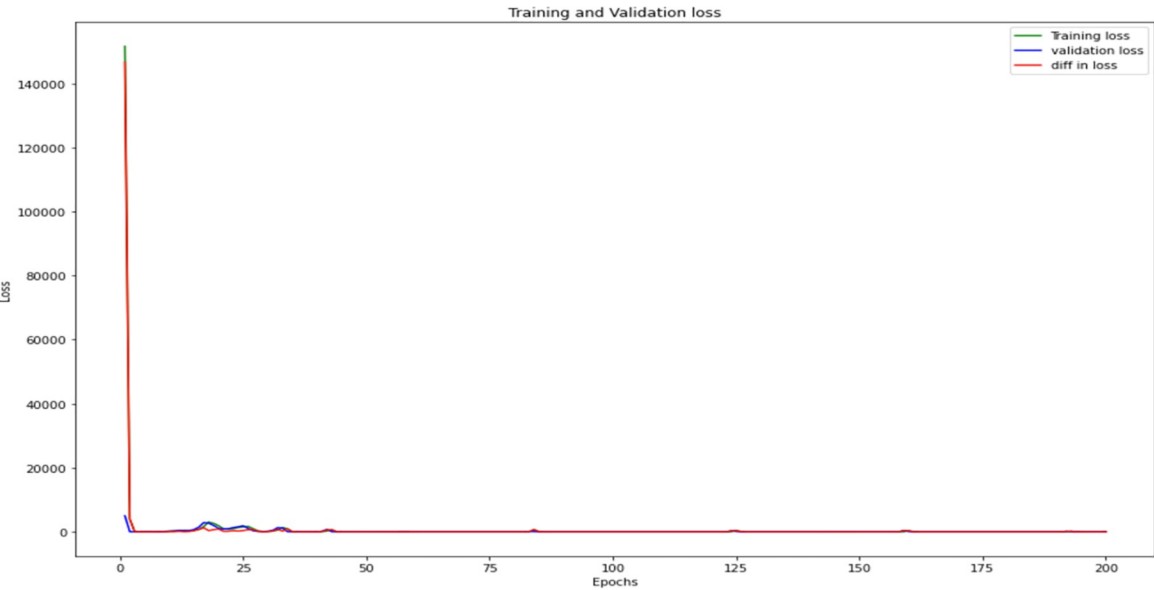

**Fig 13. Losses on each epoch and its difference.**

two losses was rather significant as shown in Fig 13, where the discrepancy steadily decreased to zero. The model's final loss calculation was 0.437, indicating that even after running the model making a perfect prediction because the loss is nearly zero. It has calculated the mean absolute error, which becomes 0.448, indicating that the gap between actual and predicted values is negligible. This demonstrates that the error rate of the training and validation models is quite low.

## 4.6 Box-Jenkins method

The Box-Jenkins approach expands on the time series data acquired for the study and forecasts the economic aspects of the UK. Initially, the data is examined using the decomposition method.

Fig 14 displays the time series data illustrating the decomposition of unemployment in the UK. The data reveals a consistent downward trend in the unemployment rate over time. Notably, there are no discernible repeating patterns within a one-year period, and the residuals are minimal, indicating the absence of seasonality. However, the declining trend suggests that the data is non-stationary.

To further analyze the data, exponential moving averages, specifically the rolling mean and rolling standard deviation, are computed. The calculation of moving averages necessitates certain assumptions about the data. In this case, it is assumed that the time series is devoid of both seasonal and trend components. This assumption implies that the time series is stationary, with no evident long-term increasing or decreasing trends, and lacks consistent periodic patterns of seasonality [43].

The initial, moving average, and standard deviation of unemployment are shown in Fig 15. There is a declining trend in the rolling mean, and the standard deviation is also growing. It demonstrates that the coefficients are independent of time. The ACF and PACF graphs are generated to confirm that the dataset is a non-stationary time series. The graph illustrates the

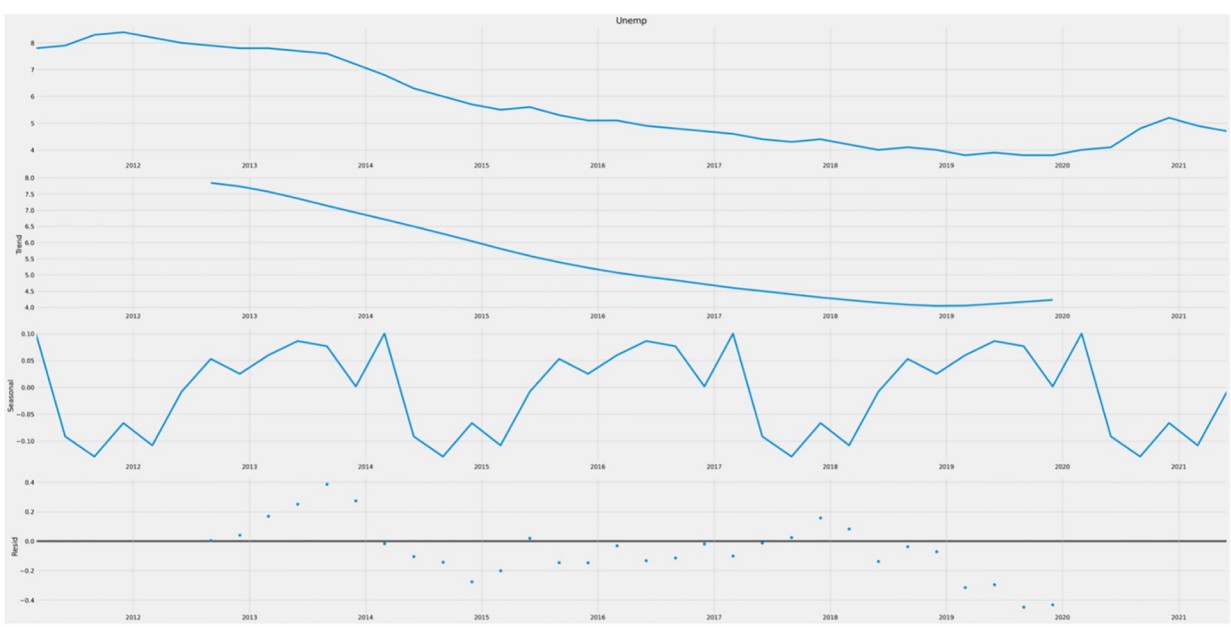

**Fig 14. Decomposition graphs.**

effect of previous values on time series values. If the time series is stationary, the ACF/PACF plots will reveal a quick termination after a small number of delays.

Figs 16 and 17 depict the ACF and PACF graphs of the UK's unemployment rates. It is feasible to conclude that the time series is randomized since the autocorrelation reduces as the number of delays are increasing. This indicates that the data are not time-dependent and are consequently non-stationary. In addition, the ADF statistical test may be used to examine the

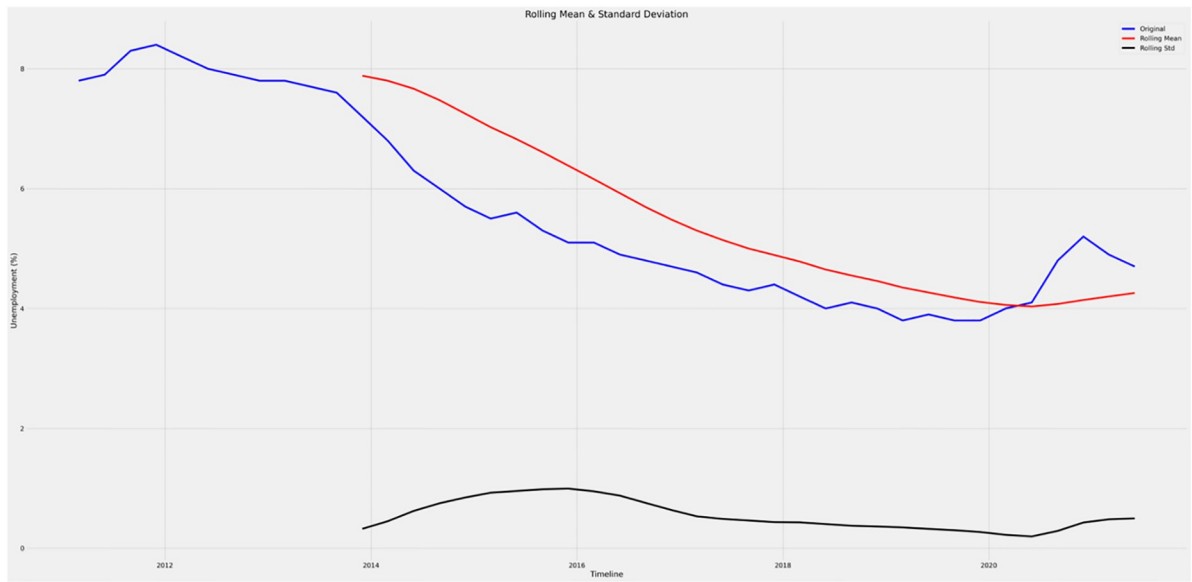

**Fig 15. Rolling-mean & standard deviation graph.**

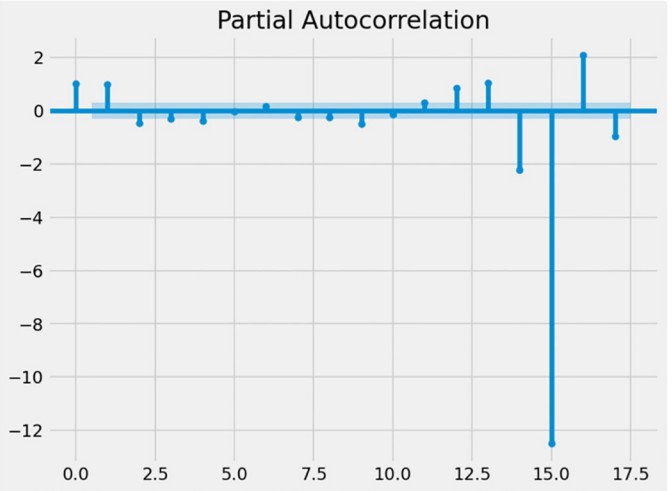

**Fig 16. PACF graph.**

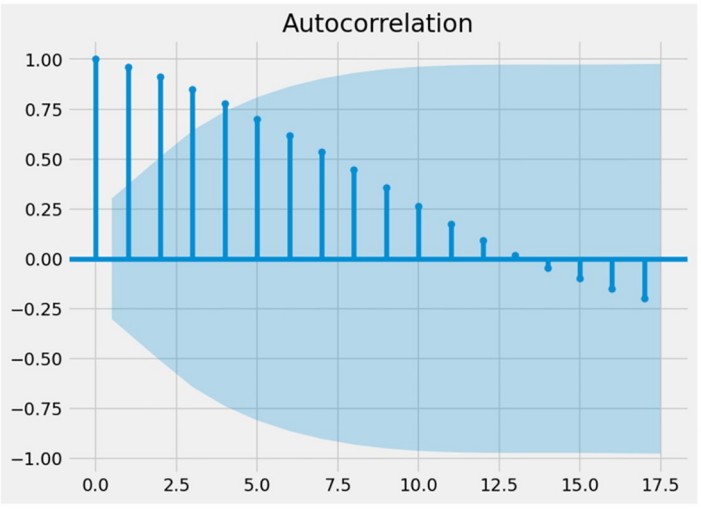

**Fig 17. ACF graph.**

stationary properties of time series data and to calculate the number of delays required to make the data stable. Since the time series has a unit root, the null hypothesis of the test is non-stationary. If the p-value is above the threshold, the null hypothesis is not rejected, and the time series is nonstationary [43].

Fig 18 depicts the residuals and data density to determine the model's validity. The chart demonstrates that residuals are between (-1,1) and there was a positive spike beyond the year 2020, indicating that unemployment grew during COVID-19 and the post-Brexit era. In addition, the density graph reveals that residuals are regularly distributed but random about 0 value. This indicates that the data can be utilised to forecast the future.

The graph in Fig 19 compares the actual and expected unemployment rates. It may be observed that anticipated values mostly overlap actual values; hence, forecasting is feasible.

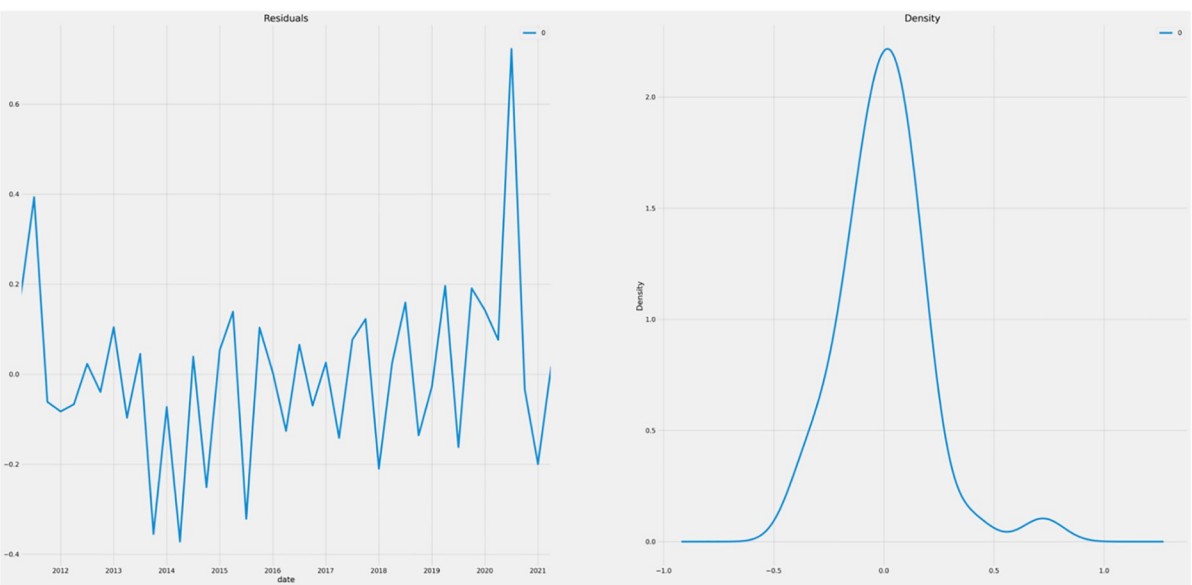

**Fig 18. Residual and density graphs.**

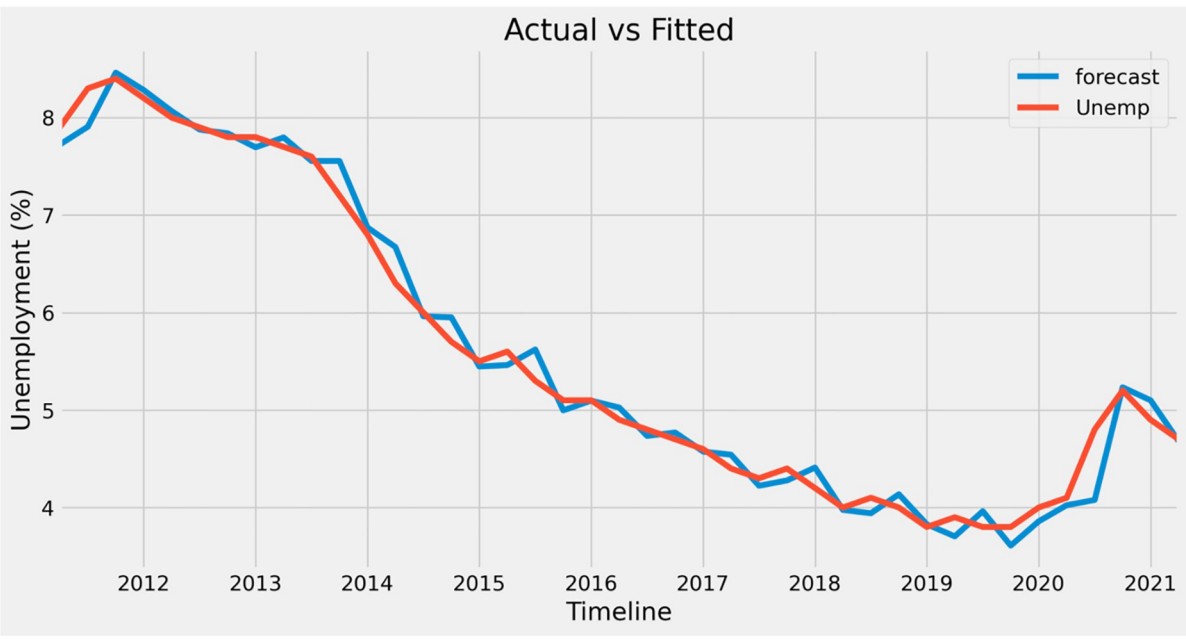

**Fig 19. Actual vs fitted graph for unemployment.**

The only exception is the year 2020, for which the predicted numbers did not match with actual values due to the unforeseeable pandemic.

It is then projected for the following five years as shown in Fig 20. Undoubtedly, unemployment rates are projected to decline, but fluctuations may arise if the economy experiences significant disruptions such as COVID-19.

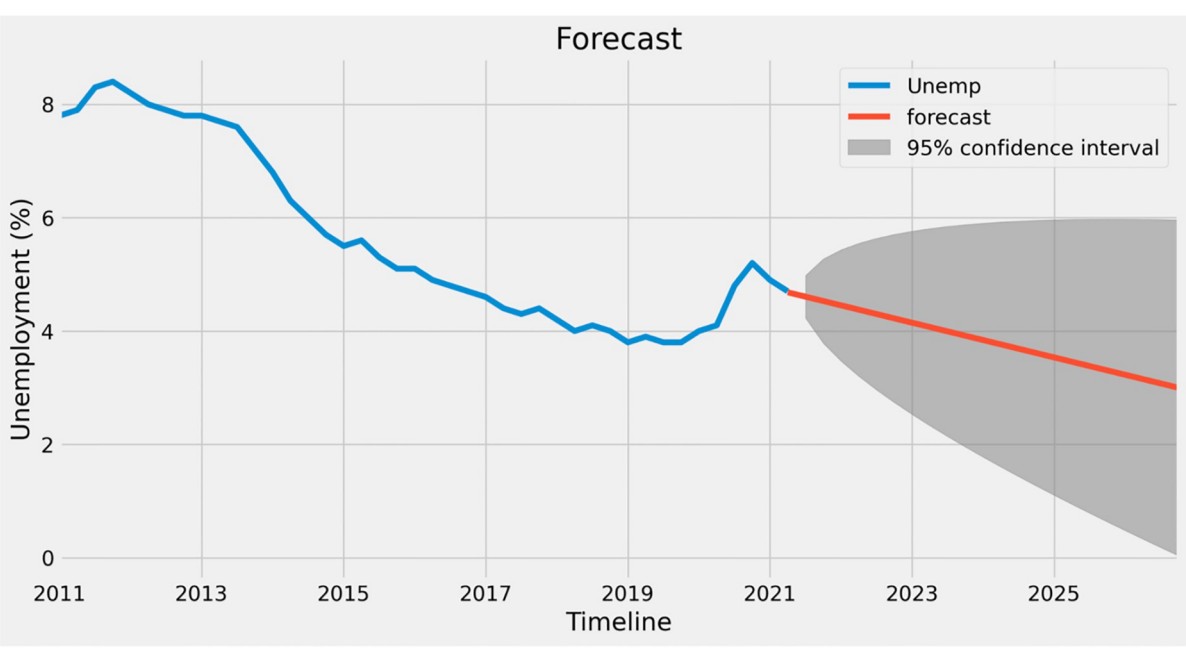

**Fig 20. Forecast graph for unemployment.**

Figs 21–23 show the forecasts of income, gross domestic product (GDP), and trade, respectively. It may be inferred that the GDP (prices) and Earnings are likely to rise in value during the next five years, while UK trade is expected to decline. This implies that even if the economy of the UK as a whole strengthens, the trade imbalance may increase and negatively affect growth in the long run.

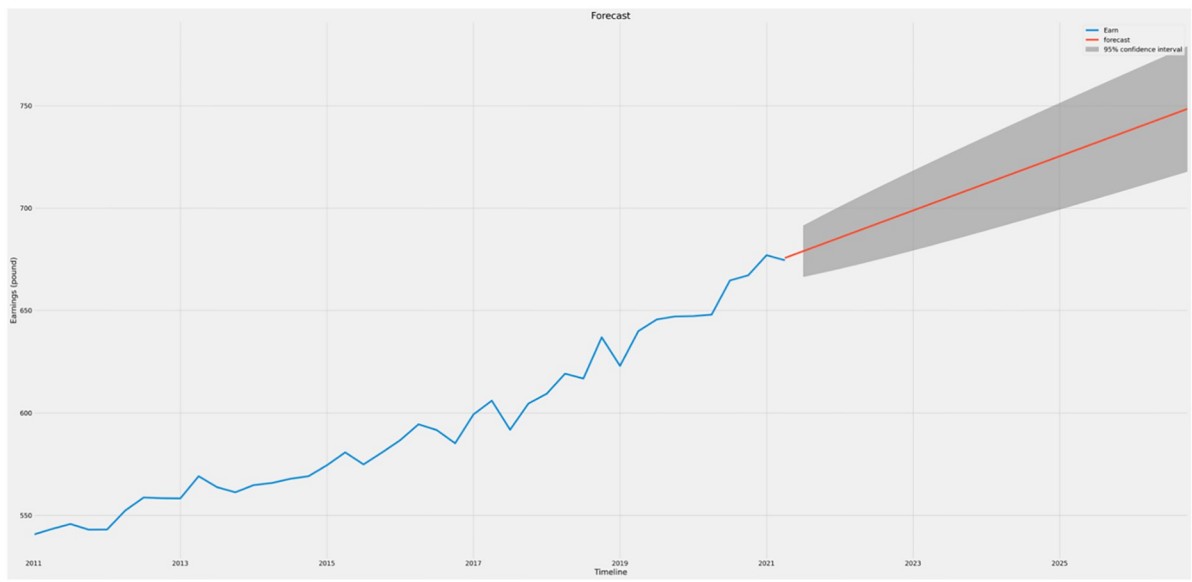

**Fig 21. Forecast graph for earnings.**

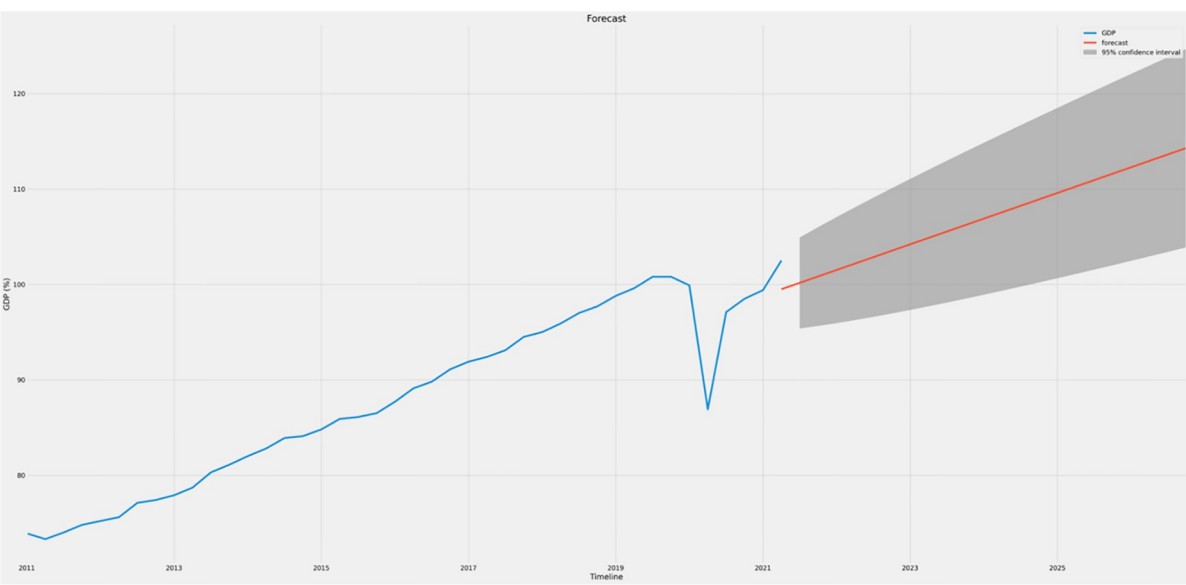

**Fig 22. Forecast graph for GDP.**

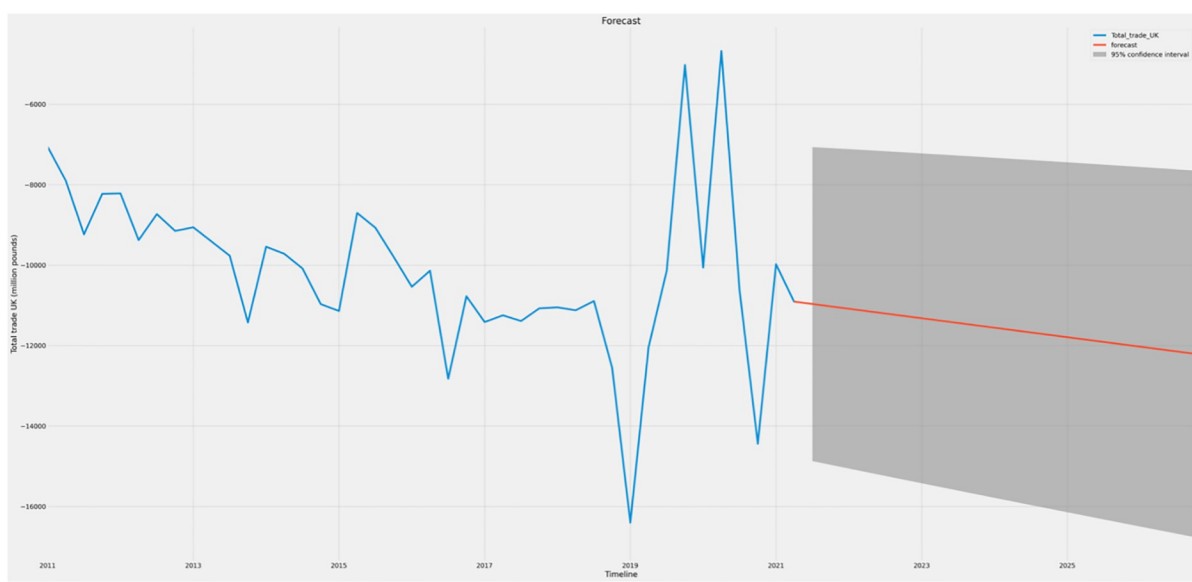

**Fig 23. Forecast graph for Trade.**

### 4.7 Public reaction

**4.7.1 Google trends.** To analyse the public's reaction to Brexit and COVID-19, the total number of web searches conducted into various categories is depicted below. Fig 24 shows the all categories of search volume for Brexit and COVID-19. It is obvious that searches for COVID-19 exceeded those for Brexit in the UK. People were most engaged in Brexit in 2016 and 2020, when the Brexit announcement was made and the transitional phase ended,

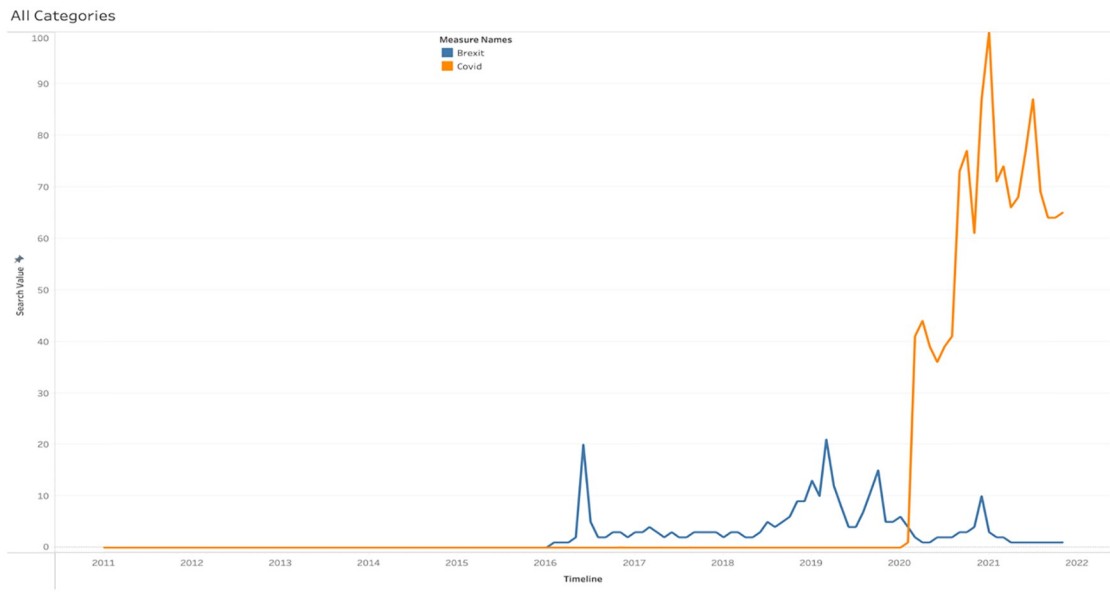

**Fig 24. Web searches of Brexit and COVID-19 under all categories.**

respectively. By the end of 2021, there are no Brexit-related searches and in contrast, COVID-19 searches began in 2020 and accelerated in 2021.

The regional searches for Brexit and COVID-19 across all categories are displayed in Table 2. This explains the popularity of Brexit and COVID-19 in various parts of the UK. England (21%) has the most Brexit-related searches, followed by Northern Ireland (19%), while Scotland (84%) has the most COVID-19-related searches, followed by Wales (82%).

Fig 25 shows the search trends for Brexit and COVID-19 in the business and industry category providing insights into the level of concern among individuals. The results indicate that during the COVID-19 period, there was an increased focus on the impact of the pandemic on the UK's businesses and industries, surpassing the levels seen in previous years. Table 3 further highlights that individuals in England expressed greater apprehension towards Brexit compared to those in other regions. Moreover, throughout the COVID-19 period, people in Scotland and Wales conducted a higher volume of searches related to UK businesses and sectors when compared to individuals in other areas.

Fig 26 displays the Brexit and COVID-19 searches under the Finance category. According to the data, Brexit had a high number of searches in the financial category in 2016 and continued until 2021. In contrast, inquiries for COVID-19 increased in 2020 and surpassed Brexit in popularity in 2016.

**Table 2. Regional popularity of Brexit and COVID-19 under all categories.**

| Region | Brexit | Covid |
|---|---|---|
| Northern Ireland | 19% | 81% |
| Wales | 18% | 82% |
| Scotland | 16% | 84% |
| England | 21% | 79% |

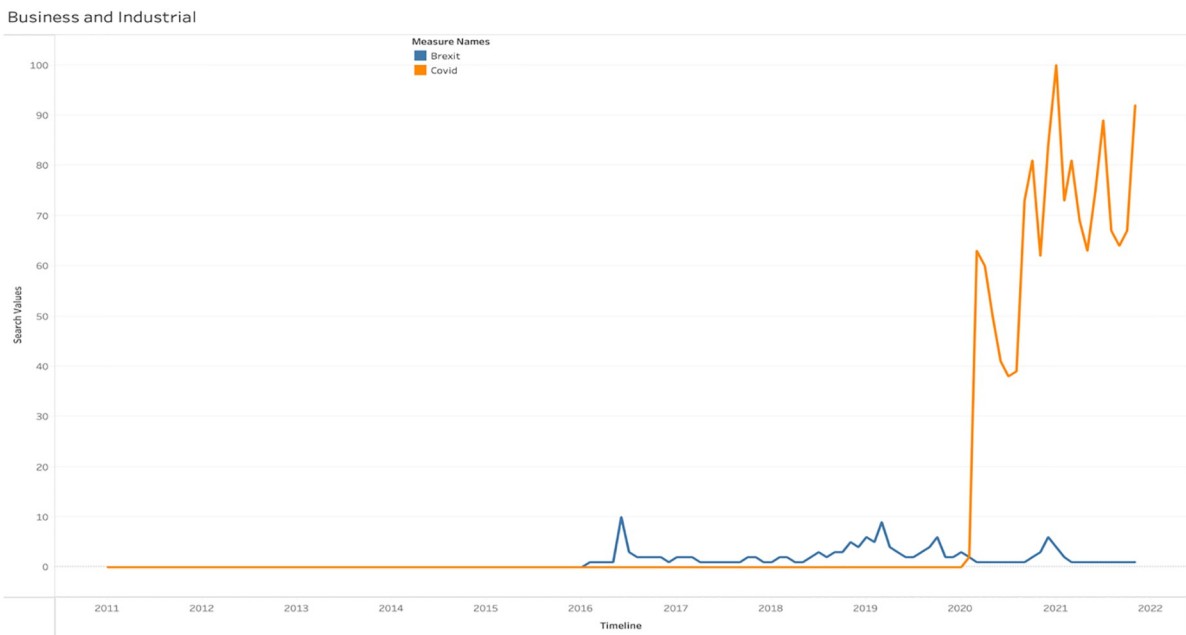

**Fig 25. Web search of the events under business and industrial category.**

The geographical searches for Brexit and COVID-19 in the finance category are displayed in Table 4. The statistics indicate that people in England conducted more searches concerning Brexit, while those in Wales conducted more searches on COVID-19.

Fig 27 depicts the Brexit and COVID-19 web search under the Jobs and Education category in the UK. Brexit searches were far lower than COVID19 searches beginning in 2020.

Table 5 displays the regional prevalence of this category. Northern Ireland had the most Brexit-related searches compared to other areas. In contrast, England and Wales had the highest number of COVID-19 searches.

**4.7.2 Twitter sentiment analysis.** In performing a Google Trend analysis, this study investigated the public's sentiment toward Brexit and COVID-19 using Twitter sentiment analysis.

Fig 28 shows the polarity of the sentiments from the tweets on Brexit. The maximum polarity appears between -0.25 and 0.25. This indicates that the majority of Brexit-related tweets were neutral. Some tweets had both negative and positive feelings, although the proportion of positive tweets was greater than negative tweets. This indicates that fewer people were entirely pleased or entirely depressed because of the incident. In addition, it can be proven that most individuals in the UK were uncertain about the consequences or impact of Brexit.

**Table 3. Regional popularity of Brexit and COVID-19 under business and industrial category.**

| Region | Brexit | Covid |
|---|---|---|
| Northern Ireland | 9% | 91% |
| Wales | 10% | 90% |
| England | 11% | 89% |
| Scotland | 10% | 90% |

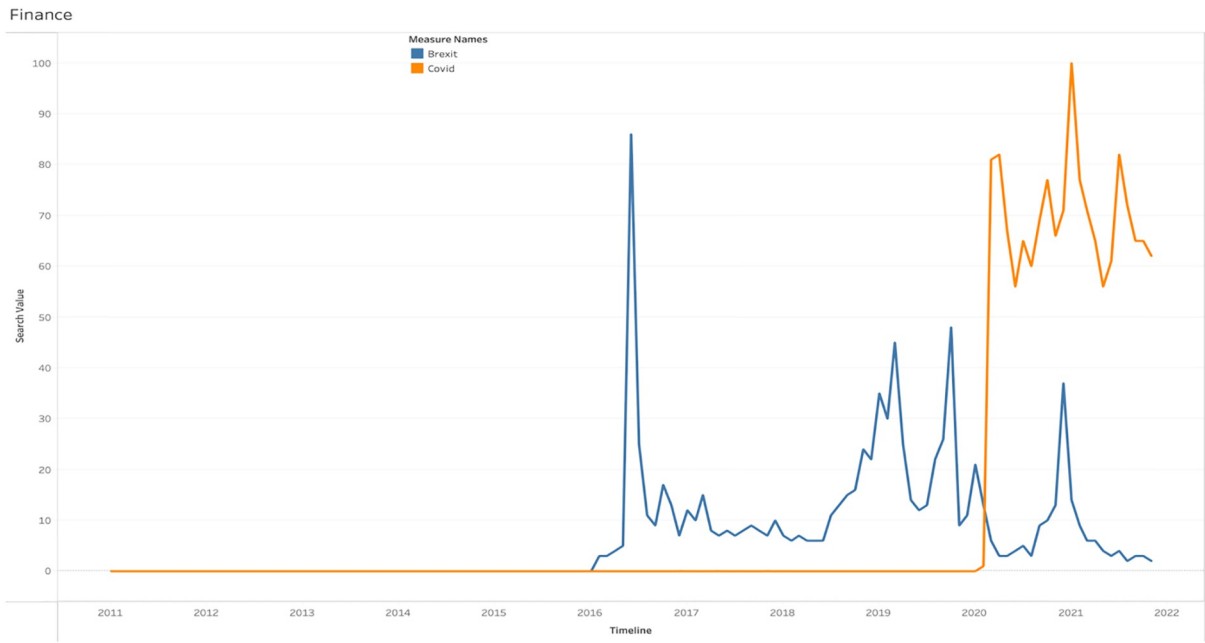

**Fig 26. Web searches of Brexit and COVID-19 under the finance category.**

Fig 29 illustrates the sensitivity of Brexit-related tweets. The majority of tweets were unrelated to Brexit, with barely a few focusing on the event. This indicates that fewer individuals tweeted about Brexit.

Fig 30 depicts the polarity of the sentiment expressed in COVID-19 tweets. The graph demonstrates that most tweets were neutral. Some of the remaining tweets were entirely positive, and the majority of them were negative. This demonstrates that individuals were depressed and had a detrimental effect on the attitudes of the British population.

Fig 31 illustrates the subjectivity of COVID-19 tweets. It is apparent that the majority of tweets are related to the epidemic since they are highly subjective. This indicates that individuals tweeted often about COVID-19 and the epidemic as a whole. The aforementioned findings demonstrate that individuals in the UK were more engaged in social media during COVID-19 than during Brexit. Also, people had considerably greater feelings about COVID-19 than they did over Brexit; as a result, the subjectivity of tweets about COVID-19 were higher than that of tweets about Brexit.

## 5 Discussion and conclusion

Understanding the economic consequences of Brexit and the COVID-19 pandemic on the United Kingdom (UK) is a complex undertaking due to their close temporal proximity,

**Table 4. Regional popularity of Brexit and COVID-19 under the finance category.**

| Region | Brexit | Covid |
|---|---|---|
| Wales | 30% | 70% |
| Northern Ireland | 35% | 65% |
| Scotland | 33% | 67% |
| England | 40% | 60% |

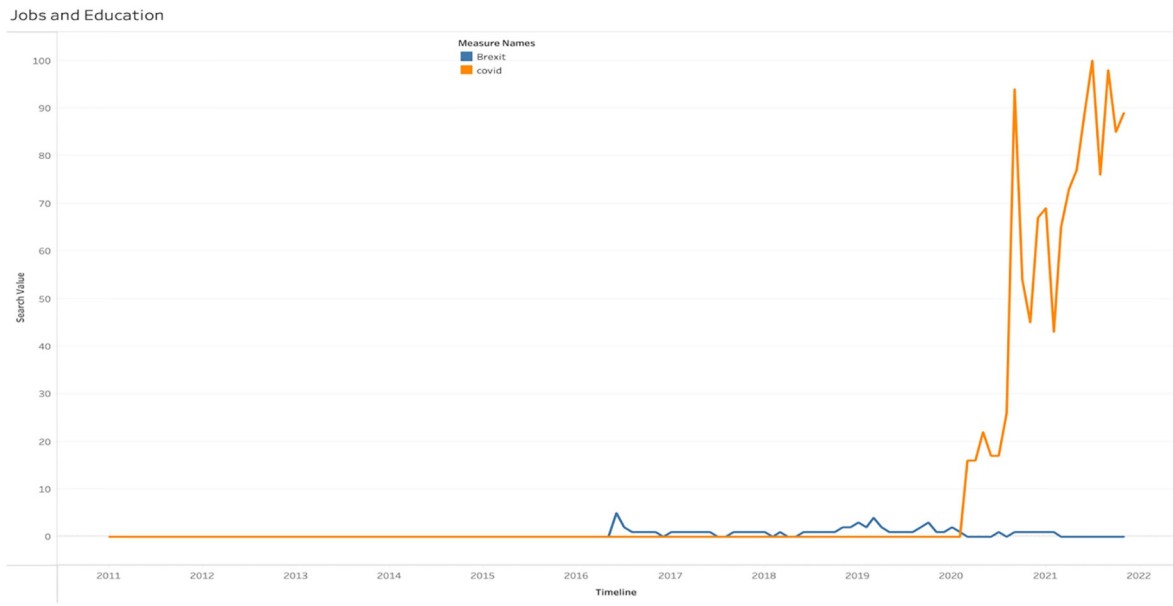

**Fig 27. Web searches of Brexit and COVID-19 under the jobs and education category.**

making it challenging to discern their individual impacts on the UK economy. Nonetheless, both events have undeniably instigated significant changes and will continue to shape the long-term economic landscape of the UK. This study aims to examine the UK economy during three distinct periods: Pre-Brexit (2011–2016), Brexit (2016–2020), and COVID-19 & Post-Brexit (2020–2021), utilizing diverse data analysis methodologies. The primary focus is to analyze key economic indicators such as the unemployment rate, GDP index, earnings, and trade across different regions and industries of the UK, providing a comprehensive understanding of the effects of Brexit and COVID-19. The findings shed light on compelling insights into the performance of the UK economy over the past decade. Visual representations of the data unveil several trends. The unemployment rate demonstrated a downward trajectory until 2020 but experienced a spike for a six-month period in 2021. Average weekly earnings exhibited a gradual increase over time, while the GDP index displayed an upward trend until 2020, followed by a decline during the COVID-19 period. Trade experienced the most significant decline post-Brexit and during the COVID-19 pandemic. Furthermore, the impact of these events varied among the four regions and twelve industries. Wales and Northern Ireland emerged as the most affected regions by both Brexit and COVID-19, whereas industries such as Accommodation, Construction, Wholesale, and Others experienced the most substantial impact on earnings and employment levels. Conversely, industries such as finance, science,

**Table 5. Regional popularity of Brexit and COVID-19 under the jobs and education category.**

| Region | Brexit | Covid |
|---|---|---|
| England | 6% | 94% |
| Wales | 6% | 94% |
| Scotland | 7% | 93% |
| Northern Ireland | 10% | 90% |

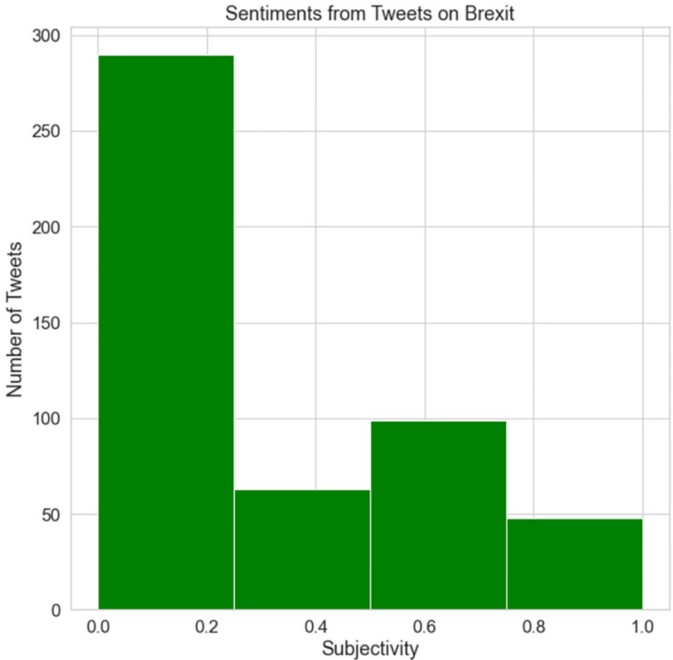

**Fig 28. The subjectivity of Brexit tweets.**

and health witnessed an increased contribution to the UK's total GDP following Brexit and COVID-19. Gender-wise, the impact was more pronounced on men than women. Among all the economic factors analyzed, trade stood out as the most affected variable in the UK. In early 2021, the macroeconomic situation in the UK revealed a discernible pattern: economic

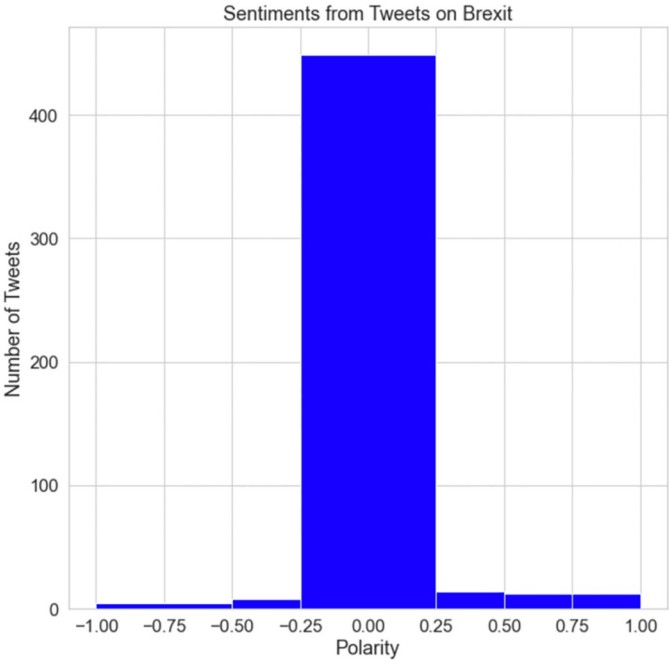

**Fig 29. The polarity of Brexit tweets.**

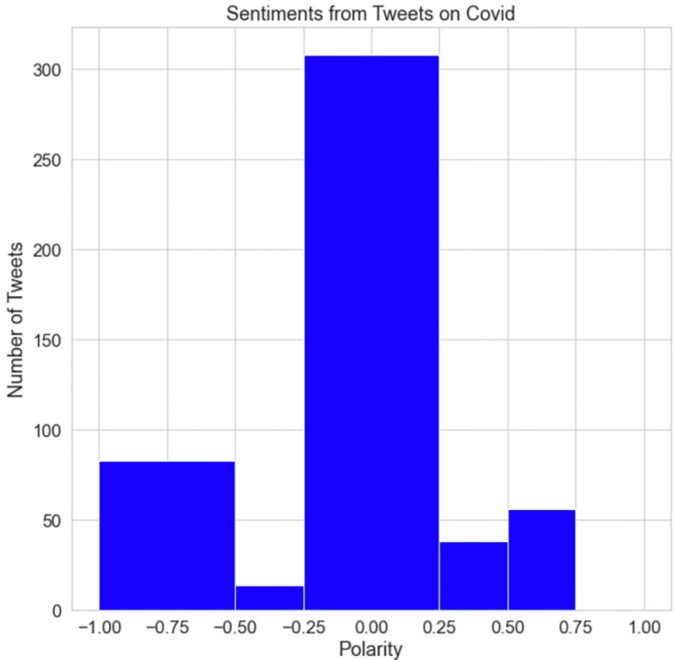

**Fig 30. The polarity of COVID-19 tweets.**

demand outpaced supply, resulting in shortages, bottlenecks, and inflation. Various industries reported a scarcity of labor, partly attributable to the global consequences of the pandemic. The findings of the neural network model indicated minimal loss between the training and validation models, indicating the model's effectiveness in predicting values with minimal error.

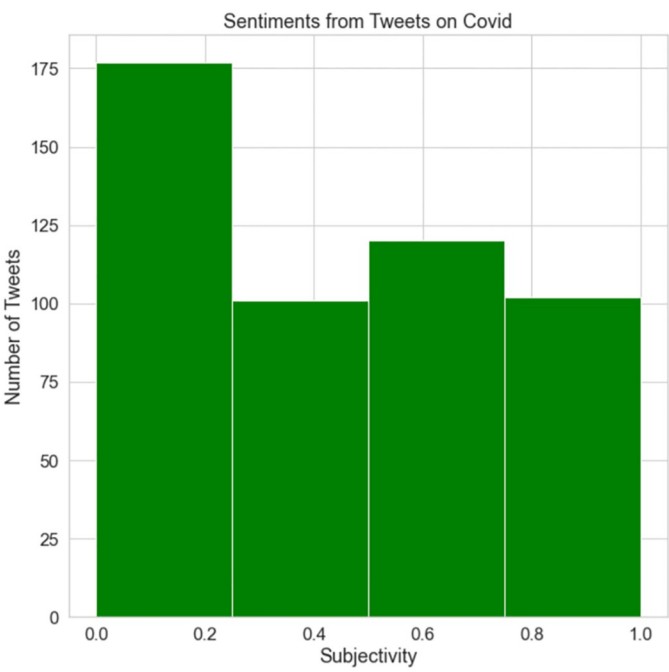

**Fig 31. The subjectivity of COVID-19 tweets.**

The Box-Jenkins analysis identified the dataset as non-stationary but with a discernible trend that could be utilized to forecast coefficients. The results of the ARIMA model projected a gradual decline in unemployment over the next five years, with the GDP index and earnings expected to continue ascending. However, trade was anticipated to decline, and net trade was forecasted to remain negative in the coming years. Finally, the results of the social media analysis confirmed that people displayed greater concern and engagement during the COVID-19 period (starting in 2020) compared to the Brexit period (starting in 2016). Individuals expressed neutrality towards Brexit and posted fewer tweets on the subject, whereas during COVID-19, people expressed unhappiness and extensively tweeted about the pandemic. It is crucial to acknowledge that these results are based on the analysis of recent data, capturing the immediate and overall impact of Brexit and COVID-19 on the UK economy. As ongoing economic fluctuations remain unpredictable, forecast predictions derived from the analysis may differ from actual outcomes. Moreover, this research relies on time-series data, which imposes limitations on the scope and accuracy of the analysis. The graphical representation of the time-series data is confined to line and bar graphs for analytical purposes. The existing literature on this topic is insufficient to validate the analysis results and support the research framework. Therefore, further work is needed to enhance the assessment of the impact of Brexit and COVID-19 on the UK economy. Continuously.

## Supporting information

**S1 File.**
(ZIP)

**S1 Dataset.**
(ZIP)

## Acknowledgments

The author would like to acknowledge the support of work from Cardiff Metropolitan University.

## Author Contributions

**Conceptualization:** Raghav Gupta.

**Data curation:** Md. Mahadi Hasan.

**Investigation:** Syed Zahurul Islam.

**Methodology:** Raghav Gupta, Md. Mahadi Hasan, Syed Zahurul Islam.

**Supervision:** Tahmina Yasmin, Jasim Uddin.

**Writing – original draft:** Raghav Gupta, Md. Mahadi Hasan, Tahmina Yasmin, Jasim Uddin.

**Writing – review & editing:** Raghav Gupta, Syed Zahurul Islam, Tahmina Yasmin, Jasim Uddin.

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
