## [Decision Letter · Decision Letter 0]

20 Feb 2023

PONE-D-22-35613Evaluating the Brexit and COVID-19’s influence on the UK Economy: A Data AnalysisPLOS ONE

Dear Dr. Uddin,

Thank you for submitting your manuscript to PLOS ONE. After careful consideration, we feel that it has merit but does not fully meet PLOS ONE’s publication criteria as it currently stands. Therefore, we invite you to submit a revised version of the manuscript that addresses the points raised during the review process.

The authors should incorporate the changes suggested by reviewers. 

We look forward to receiving your revised manuscript.

Kind regards,

Fiza Qureshi, PhD

Academic Editor

PLOS ONE

Journal Requirements:

3. In your Methods section, please include additional information about your dataset and ensure that you have included a statement specifying whether the collection and analysis method complied with the terms and conditions for the source of the data.

"NO"

Reviewers' comments:

Reviewer's Responses to Questions

**Comments to the Author**

1. Is the manuscript technically sound, and do the data support the conclusions?

Reviewer #1: Yes

Reviewer #2: Yes

2. Has the statistical analysis been performed appropriately and rigorously? 

Reviewer #1: Yes

Reviewer #2: Yes

3. Have the authors made all data underlying the findings in their manuscript fully available?

Reviewer #1: Yes

Reviewer #2: Yes

4. Is the manuscript presented in an intelligible fashion and written in standard English?

Reviewer #1: Yes

Reviewer #2: Yes

5. Review Comments to the Author

Reviewer #1: This paper examines Brexit and COVID-19’s influence on the UK economy. The idea is interesting but the paper could be improved as I will explain below.

First, there is a need to carefully proofread the paper. For example, “last” should be “Last” in the opening of the Introduction.

Second, an early study about Brexit in 2018 should be acknowledged so that readers understand when it started, which could also be explained theoretically through the lens of de-internationalization:

Exiting supranational unions and the corresponding impact on tourism: Some insights from a rejoinder to Brexit. Current Issues in Tourism, 21(9), 970-974.

A general theory of de‐internationalization. Global Business and Organizational Excellence, 42(2), 9-15.

De‐internationalization: An organizational institutionalism perspective. Global Business and Organizational Excellence, 42(3), 58-73.

Third, some lessons from the COVID-19 pandemic should be added:

History, lessons, and ways forward from the COVID-19 pandemic. International Journal of Quality and Innovation, 5(2), 101-108.

Fourth, some insights on how COVID-19 has impacted on different economies should be added and why a UK focus would add to the field:

The economic impact of a global pandemic on the tourism economy: The case of COVID-19 and Macao’s destination-and gambling-dependent economy. Current Issues in Tourism, 25(8), 1258-1269.

The quarantine economy: The case of COVID-19 and Malaysia. In COVID-19, Business, and Economy in Malaysia (pp. 3-23). New York: Routledge.

Fifth, the conditional approach that is taken to review the impact of Brexit and COVID-19 (i.e., before and after) should be clearly indicated and supported:

Conditional recipes for predicting impacts and prescribing solutions for externalities: The case of COVID-19 and tourism. Tourism Recreation Research, 46(2), 314-318.

Sixth, since trends and sentiment analysis appear to be of interest, I reckon it would be good to also add some bibliometric insights on Brexit versus COVID-19 research in the UK. Some ideas about bibliometric analysis and its contributions could be found in:

How to conduct a bibliometric analysis: An overview and guidelines. Journal of Business Research, 133, 285–296.

Guidelines for advancing theory and practice through bibliometric research. Journal of Business Research, 148, 101–115.

Seventh, some insights about the challenges of COVID-19 with regard to the business and economy could be found:

COVID-19, Business, and economy in Malaysia: retrospective and prospective perspectives. New York: Routledge.

Transformative marketing in the new normal: A novel practice-scholarly integrative review of business-to-business marketing mix challenges, opportunities, and solutions. Journal of Business Research.

Finally, some ideas on future research could be added to enrich the paper. Some ideas could be found below:

Ushering a new era of Global Business and Organizational Excellence: Taking a leaf out of recent trends in the new normal. Global Business and Organizational Excellence, 41(5), 5-13.

What is at stake in a war? A prospective evaluation of the Ukraine and Russia conflict for business and society. Global Business and Organizational Excellence, 41(6), 23-36.

I hope you will find these comments useful to improve the paper.

Good luck and all the best!

Reviewer #2: Over, the study covers contemporary issue and uses multiple data techniques to analyze different aspects of the study. However, some minor corrections are needed as mentioned below:

1. Abstract needs to be rewrite in more cohesive and concise manner.

2. There is need to provide more rationale regarding the choices of economic factors studied in the paper. Why some other factors like inflation, investment are not being studied.

3. There could be some comparative discussion needs to be included in terms of either Brexit influenced economy more or COVID-19 ?

4. There is need to provide rationale regarding the selection of data analysis techniques. Does these techniques enrich study findings.

5. Thorough proofreading of the manuscript is required.

6. PLOS authors have the option to publish the peer review history of their article (what does this mean?). If published, this will include your full peer review and any attached files.

Reviewer #1: No

Reviewer #2: **Yes: **Sobia Shafaq Shah

---

## [Author Response · Author response to Decision Letter 0]

12 Apr 2023

Please find the attached following files dated: 05/04/2023

1) Explicit Statement

2) Manuscript 05042023

3) S2 (zip file)-> Data coding files

4) Source file_PLOS-Brexit -> Overleaf/latex manuscript file

---

## [Decision Letter · Decision Letter 1]

27 Apr 2023

PONE-D-22-35613R1Evaluating the Brexit and COVID-19’s influence on the UK Economy: A Data AnalysisPLOS ONE

Dear Dr. Uddin,

Thank you for submitting your manuscript to PLOS ONE. After careful consideration, we feel that it has merit but does not fully meet PLOS ONE’s publication criteria as it currently stands. Therefore, we invite you to submit a revised version of the manuscript that addresses the points raised during the review process.

Dear Authors, Please address the reviewer 1 comments. 

We look forward to receiving your revised manuscript.

Kind regards,

Fiza Qureshi, PhD

Academic Editor

PLOS ONE

Journal Requirements:

Reviewers' comments:

Reviewer's Responses to Questions

**Comments to the Author**

1. If the authors have adequately addressed your comments raised in a previous round of review and you feel that this manuscript is now acceptable for publication, you may indicate that here to bypass the “Comments to the Author” section, enter your conflict of interest statement in the “Confidential to Editor” section, and submit your "Accept" recommendation.

Reviewer #1: (No Response)

Reviewer #2: All comments have been addressed

2. Is the manuscript technically sound, and do the data support the conclusions?

Reviewer #1: Yes

Reviewer #2: Yes

3. Has the statistical analysis been performed appropriately and rigorously? 

Reviewer #1: Yes

Reviewer #2: Yes

4. Have the authors made all data underlying the findings in their manuscript fully available?

Reviewer #1: Yes

Reviewer #2: Yes

5. Is the manuscript presented in an intelligible fashion and written in standard English?

Reviewer #1: Yes

Reviewer #2: Yes

6. Review Comments to the Author

Reviewer #1: I do not see the reviewer comments being addressed (only the editors’ comments were addressed), so I’m returning with the previous comments for the authors to work on.

his paper examines Brexit and COVID-19’s influence on the UK economy. The idea is interesting but the paper could be improved as I will explain below.

First, there is a need to carefully proofread the paper. For example, “last” should be “Last” in the opening of the Introduction.

Second, an early study about Brexit in 2018 should be acknowledged so that readers understand when it started, which could also be explained theoretically through the lens of de-internationalization:

Exiting supranational unions and the corresponding impact on tourism: Some insights from a rejoinder to Brexit. Current Issues in Tourism, 21(9), 970-974.

A general theory of de‐internationalization. Global Business and Organizational Excellence, 42(2), 9-15.

De‐internationalization: An organizational institutionalism perspective. Global Business and Organizational Excellence, 42(3), 58-73.

Third, some lessons from the COVID-19 pandemic should be added:

History, lessons, and ways forward from the COVID-19 pandemic. International Journal of Quality and Innovation, 5(2), 101-108.

Fourth, some insights on how COVID-19 has impacted on different economies should be added and why a UK focus would add to the field:

The economic impact of a global pandemic on the tourism economy: The case of COVID-19 and Macao’s destination-and gambling-dependent economy. Current Issues in Tourism, 25(8), 1258-1269.

The quarantine economy: The case of COVID-19 and Malaysia. In COVID-19, Business, and Economy in Malaysia (pp. 3-23). New York: Routledge.

Fifth, the conditional approach that is taken to review the impact of Brexit and COVID-19 (i.e., before and after) should be clearly indicated and supported:

Conditional recipes for predicting impacts and prescribing solutions for externalities: The case of COVID-19 and tourism. Tourism Recreation Research, 46(2), 314-318.

Sixth, since trends and sentiment analysis appear to be of interest, I reckon it would be good to also add some bibliometric insights on Brexit versus COVID-19 research in the UK. Some ideas about bibliometric analysis and its contributions could be found in:

How to conduct a bibliometric analysis: An overview and guidelines. Journal of Business Research, 133, 285–296.

Guidelines for advancing theory and practice through bibliometric research. Journal of Business Research, 148, 101–115.

Seventh, some insights about the challenges of COVID-19 with regard to the business and economy could be found:

COVID-19, Business, and economy in Malaysia: retrospective and prospective perspectives. New York: Routledge.

Transformative marketing in the new normal: A novel practice-scholarly integrative review of business-to-business marketing mix challenges, opportunities, and solutions. Journal of Business Research.

Finally, some ideas on future research could be added to enrich the paper. Some ideas could be found below:

Ushering a new era of Global Business and Organizational Excellence: Taking a leaf out of recent trends in the new normal. Global Business and Organizational Excellence, 41(5), 5-13.

What is at stake in a war? A prospective evaluation of the Ukraine and Russia conflict for business and society. Global Business and Organizational Excellence, 41(6), 23-36.

I hope you will find these comments useful to improve the paper.

Good luck and all the best!

Reviewer #2: The authors have adequately responded to all queries and have provided sufficient details in methodology section to support study findings. Overall, study meets the standard criteria to get publish in PLOS ONE.

7. PLOS authors have the option to publish the peer review history of their article (what does this mean?). If published, this will include your full peer review and any attached files.

Reviewer #1: No

Reviewer #2: **Yes: **Sobia Shafaq Shah

---

## [Author Response · Author response to Decision Letter 1]

12 May 2023

Attached the following update version new files:

1. Original Manuscript

2. Tracking Manuscript

3. Latex source files (zipped)

4. Response to the Reviewer (Latest 13/05/2023)

5. Response to the Reviewer (older version)

---

## [Decision Letter · Decision Letter 2]

5 Jun 2023

Evaluating the Brexit and COVID-19’s influence on the UK Economy: A Data Analysis

PONE-D-22-35613R2

Dear Dr. Uddin,

We’re pleased to inform you that your manuscript has been judged scientifically suitable for publication and will be formally accepted for publication once it meets all outstanding technical requirements.

Kind regards,

Umer Shahzad, PhD

Academic Editor

PLOS ONE

Additional Editor Comments (optional):

Reviewers' comments:

Reviewer's Responses to Questions

**Comments to the Author**

1. If the authors have adequately addressed your comments raised in a previous round of review and you feel that this manuscript is now acceptable for publication, you may indicate that here to bypass the “Comments to the Author” section, enter your conflict of interest statement in the “Confidential to Editor” section, and submit your "Accept" recommendation.

Reviewer #1: All comments have been addressed

Reviewer #2: All comments have been addressed

2. Is the manuscript technically sound, and do the data support the conclusions?

Reviewer #1: Yes

Reviewer #2: Yes

3. Has the statistical analysis been performed appropriately and rigorously? 

Reviewer #1: Yes

Reviewer #2: Yes

4. Have the authors made all data underlying the findings in their manuscript fully available?

Reviewer #1: Yes

Reviewer #2: Yes

5. Is the manuscript presented in an intelligible fashion and written in standard English?

Reviewer #1: Yes

Reviewer #2: Yes

6. Review Comments to the Author

Reviewer #1: Thank you for revising the paper once again, which is clearly much better than the previous version.

I am happy to recommend this version of the paper for acceptance. Congratulations!

Reviewer #2: The authors have paid adequate attention to the revisions suggested and have revised manuscript accordingly. The issues of lack of consistency and some ambiguities in earlier version of manuscript are well sorted out.

7. PLOS authors have the option to publish the peer review history of their article (what does this mean?). If published, this will include your full peer review and any attached files.

Reviewer #1: No

Reviewer #2: **Yes: **Sobia Shafaq Shah

---

## [Editor Report · Acceptance letter]

7 Jun 2023

PONE-D-22-35613R2 

Evaluating the Brexit and COVID-19’s influence on the UK Economy: A Data Analysis 

Dear Dr. Uddin:

I'm pleased to inform you that your manuscript has been deemed suitable for publication in PLOS ONE. Congratulations! Your manuscript is now with our production department. 

Kind regards, 

on behalf of

Dr. Umer Shahzad 

Academic Editor

PLOS ONE